# Efficient Zero-Shot Inpainting with Decoupled Diffusion Guidance

**Badr Moufad[1,*]  Navid Bagheri Shouraki[1,5,7]  Alain Oliviero Durmus[1]**
**Thomas Hirtz[6]  Eric Moulines[3,4]  Jimmy Olsson[8]  Yazid Janati[2,3,*]**

[1]CMAP, Ecole Polytechnique  [2]Institute of Foundation Models  [3]MBZUAI  [4]EPITA
[5]Sorbonne University  [6]Lagrange Mathematics and Computing Research Center
[7]EPITA Research Lab  [8]KTH Royal Institute of Technology

## Abstract

Diffusion models have emerged as powerful priors for image editing tasks such as inpainting and local modification, where the objective is to generate realistic content that remains consistent with observed regions. In particular, zero-shot approaches that leverage a pretrained diffusion model, without any retraining, have been shown to achieve highly effective reconstructions. However, state-of-the-art zero-shot methods typically rely on a sequence of surrogate likelihood functions, whose scores are used as proxies for the ideal score. This procedure however requires vector-Jacobian products through the denoiser at every reverse step, introducing significant memory and runtime overhead. To address this issue, we propose a new likelihood surrogate that yields simple and efficient to sample Gaussian posterior transitions, sidestepping the backpropagation through the denoiser network. Our extensive experiments show that our method achieves strong observation consistency compared with fine-tuned baselines and produces coherent, high-quality reconstructions, all while significantly reducing inference cost. Code is available at https://github.com/YazidJanati/ding.

## 1 Introduction

We focus on *inpainting problems* in computer vision, which play a central role in applications ranging from photo restoration to content creation and interactive design. Given an image with prescribed missing pixels, the objective is to generate a semantically coherent completion while ensuring strict consistency with the observed region. The importance of this task has motivated extensive research, spanning both classical approaches and, more recently, generative modeling with diffusion models (Rombach et al., 2022; Esser et al., 2024; Batifol et al., 2025; Wu et al., 2025).

To address this problem, two main diffusion-based approaches have been popularized. The first relies on training *conditional diffusion models* tailored to a specific editing setup. These models directly approximate the conditional distribution of interest (Saharia et al., 2022; Wang et al., 2023a; Kawar et al., 2023; Huang et al., 2025) and take as side inputs additional information such as a mask, a text prompt, or reference pixels (Saharia et al., 2022; Wang et al., 2023a; Kawar et al., 2023; Huang et al., 2025). An alternative approach, which has recently attracted growing attention, is *zero-shot image editing*, requiring no extra training or fine-tuning. In this formulation, the task is cast as a Bayesian inverse problem: the pre-trained diffusion model serves as a prior, while a likelihood term enforces fidelity to the observations, and the resulting posterior distribution defines the reconstructions (Song & Ermon, 2019; Song et al., 2021b; Kadkhodaie & Simoncelli, 2020; Kawar et al., 2022; Lugmayr et al., 2022; Avrahami et al., 2022; Chung et al., 2023; Mardani et al., 2024; Rout et al., 2024a). Sampling from this posterior is achieved by approximating the score functions associated with the diffusion model adapted to this distribution. This *plug-and-play* paradigm has been investigated across a variety

---

*Authors contributed equally
Correspondence: {badr.moufad@polytechnique.edu}, {yazid.janati@mbzuai.ac.ae}

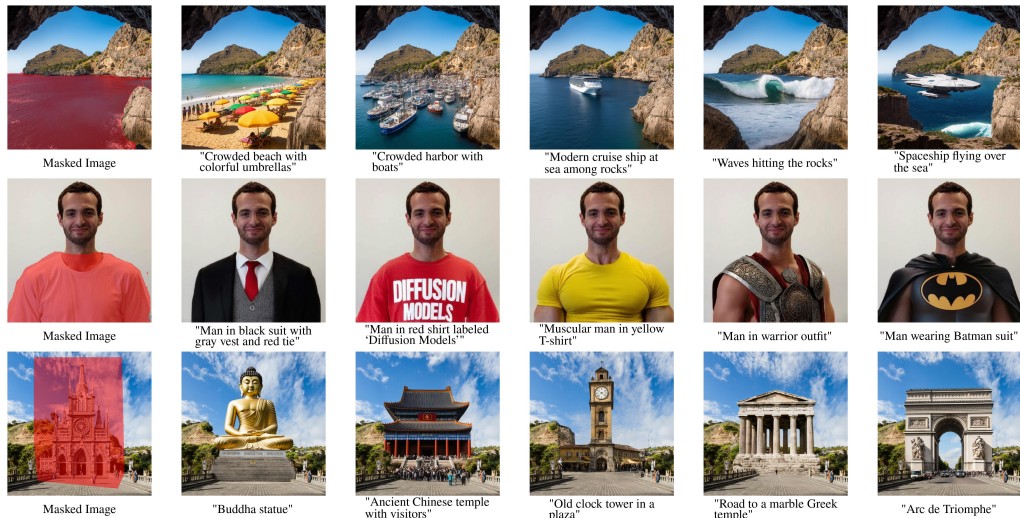

Figure 1: Zero-shot inpainting edits generated by DING (50 NFEs) for different masking patterns using Stable Diffusion 3.5 (medium). Given masked inputs (left column), the model fills the missing regions according to diverse textual prompts.

of inverse problems, from image restoration to scientific imaging, and has demonstrated strong editing performance without task-specific training.

While current zero-shot methods are appealing, they face a critical practical limitation. Implementations of strong zero-shot posterior sampling with diffusion priors typically rely on the twisting function proposed by Ho et al. (2022); Chung et al. (2023); Song et al. (2023a), which corresponds to the likelihood evaluated at the denoiser's output given the observation. Simulating the corresponding reverse diffusion process requires computing gradients of the denoiser with respect to its input. This in turn entails repeated backpropagation through the denoiser network and costly vector–Jacobian product (VJP) evaluations. This makes such methods computationally demanding, memory intensive, and often slower than training a dedicated conditional model.

**Contributions.** We propose a new *VJP-free* framework for zero-shot inpainting with a pre-trained diffusion prior. Our key idea is to approximate the intractable twisted posterior-sampling transitions by a closed-form mixture distribution that can be sampled exactly, thereby eliminating the need for VJP evaluations and backpropagation through the denoiser. Concretely, we modify the twisting function of Ho et al. (2022); Chung et al. (2023) so that it evaluates the denoiser at an independent draw from the pretrained transition. This decoupling breaks the dependency between the denoiser and the arguments of the transition density. As a result, our method provides posterior transitions that can be sampled efficiently for zero-shot inpainting with latent diffusion models. We demonstrate through extensive experiments on Stable Diffusion (SD) 3.5 that our method, coined DECOUPLED INPAINTING GUIDANCE (DING), consistently outperforms state-of-the-art guidance methods under low NFE budgets. It achieves, across three benchmarks, the best trade-off between fidelity to the visible content and realism of the reconstructions, while being both faster and more memory-efficient than competing approaches. Remarkably, even without any task-specific fine-tuning, it outperforms an SD 3 model that has been fine-tuned for image editing, confirming the effectiveness and practicality of our framework.

## 2 BACKGROUND

**Diffusion models** Denoising diffusion models (DDMs) (Sohl-Dickstein et al., 2015; Song & Ermon, 2019; Ho et al., 2020) define a generative process for a data distribution $p_0$ by constructing a continuous path $(p_t)_{t \in [0,1]}$ of distributions between $p_0$ and $p_1 := \mathcal{N}(0, I_d)$. More precisely,

$p_t = \mathrm{Law}(X_t)$, where

$$X_t = \alpha_t X_0 + \sigma_t X_1 \,, \quad X_0 \sim p_0 \,, \quad X_1 \sim p_1 \,. \tag{2.1}$$

Here $X_0$ and $X_1$ are supposed to be independent and $(\alpha_t)_{t \in [0,1]}$ and $(\sigma_t)_{t \in [0,1]}$ are deterministic, non-increasing and non-decreasing, respectively, schedules with boundary conditions $(\alpha_0, \sigma_0) :=$ $(1,0)$ and $(\alpha_1, \sigma_1) := (0,1)$. Typical choices include the *variance-preserving schedule*, satisfying $\alpha_t^2 + \sigma_t^2 = 1$ (Ho et al., 2020; Dhariwal & Nichol, 2021), and the *linear schedule*, defined by $(\alpha_t, \sigma_t) = (1-t, t)$ (Lipman et al., 2023; Esser et al., 2024; Gao et al., 2024). The path $(p_t)_{t \in [0,1]}$ defines an interpolation that gradually transforms the clean data distribution $p_0$ into the Gaussian reference distribution $p_1$. To generate new samples, DDMs simulate a time-reversed Markov chain. Given a decreasing sequence $(t_k)_{k=0}^K$ of time steps with $t_K = 1$ and $t_0 = 0$, reverse transitions are iteratively applied to map a sample from $p_{t_{k+1}}$ to one from $p_{t_k}$, thereby progressively denoising until convergence to the clean distribution $p_0$.

The DDIM framework (Song et al., 2021a) introduces a general family of reverse transitions for denoising diffusion models. It defines a new schedule $(\eta_t)_{t \in [0,1]}$, satisfying $\eta_t \leq \sigma_t$ for all $t \in [0,1]$, along with a family of transition densities given for $s < t$ by

$$p_{s|t}^\eta(\mathbf{x}_s | \mathbf{x}_t) = \mathbb{E}\left[ q_{s|0,1}^\eta(\mathbf{x}_s | X_0, X_1) \,\Big|\, X_t = \mathbf{x}_t \right] \,, \tag{2.2}$$

where $q_{s|0,1}^\eta(\mathbf{x}_s | \mathbf{x}_0, \mathbf{x}_1) := \mathrm{N}(\mathbf{x}_s; \alpha_s \mathbf{x}_0 + \sqrt{\sigma_s^2 - \eta_s^2}\, \mathbf{x}_1, \eta_s^2 \mathrm{I})$ and the random variables $(X_0, X_t, X_1)$ are defined as in (2.1). By construction, this family satisfies the marginalization property $p_s(\mathbf{x}_s) = \int p_{s|t}^\eta(\mathbf{x}_s | \mathbf{x}_t)\, p_t(\mathbf{x}_t)\, \mathrm{d}\mathbf{x}_t$ (Song et al., 2021a, Appendix B). Thus, $(p_{t_k|t_{k+1}}^\eta)_{k=0}^{K-1}$ defines a consistent set of reverse transitions, enabling stepwise sampling from the sequence $(p_{t_k})_{k=0}^K$. In practice, however, these transitions are intractable. A common approximation is to replace $X_0$ and $X_1$ in (2.2) by their conditional expectations (Ho et al., 2020; Song et al., 2021a). More precisely, let $\hat{\mathbf{x}}_0^\theta(\cdot, t)$ denote a parametric estimator of $\hat{\mathbf{x}}_0(\mathbf{x}_t, t) := \mathbb{E}[X_0 \mid X_t = \mathbf{x}_t]$. Since $\mathbb{E}[X_1 | X_t = \mathbf{x}_t] = (\mathbf{x}_t - \alpha_t \hat{\mathbf{x}}_0(\mathbf{x}_t, t))/\sigma_t$, we set $\hat{\mathbf{x}}_1^\theta(\mathbf{x}_t, t) := (\mathbf{x}_t - \alpha_t \hat{\mathbf{x}}_0^\theta(\mathbf{x}_t, t))/\sigma_t$. Then the parametric model proposed by Ho et al. (2020); Song et al. (2021a) corresponds to approximating each $p_{t_k|t_{k+1}}^\eta$ by

$$p_{t_k|t_{k+1}}^{\eta,\theta}(\mathbf{x}_{t_k} | \mathbf{x}_{t_{k+1}}) := q_{t_k|0,1}^\eta(\mathbf{x}_{t_k} | \hat{\mathbf{x}}_0^\theta(\mathbf{x}_{t_{k+1}}, t_{k+1}), \hat{\mathbf{x}}_1^\theta(\mathbf{x}_{t_{k+1}}, t_{k+1})) \,. \tag{2.3}$$

For $k = 0$, $p_{0|t_1}^{\eta,\theta}(\cdot | \mathbf{x}_{t_1})$ is simply defined as the Dirac mass at $\hat{\mathbf{x}}_0^\theta(\mathbf{x}_{t_1}, t_1)$. In the rest of the paper we omit the superscript $\eta$ when there is no ambiguity.

**Image editing.** In this work, we address the task of image editing via inpainting. We assume access to some reference image $\mathbf{x}_* \in \mathbb{R}^d$ that must be modified while remaining consistent with a prescribed set of observed pixels. Let $\mathbf{m} \subset \{1, \ldots, d\}$ denote the index set of missing (masked) pixels, and let $\overline{\mathbf{m}} = \{1, \ldots, d\} \setminus \mathbf{m}$ be the index set of observed (unmasked) pixels, with cardinality $|\overline{\mathbf{m}}| = d_{\mathbf{y}}$. For any $\mathbf{x} \in \mathbb{R}^d$ and $\mathbf{i} \subset \{1, \ldots, d\}$, we denote by $\mathbf{x}[\mathbf{i}] \in \mathbb{R}^{|\mathbf{i}|}$ the subvector formed by the components of $\mathbf{x}$ with indices $\mathbf{i}$. The observation is thus given by $\mathbf{y} := \mathbf{x}_*[\overline{\mathbf{m}}]$, and the objective is to synthesize a reconstruction $\hat{\mathbf{x}}$ such that $\hat{\mathbf{x}}[\overline{\mathbf{m}}] \approx \mathbf{y}$ while generating the missing region $\hat{\mathbf{x}}[\mathbf{m}]$ in a realistic and semantically coherent manner with respect to the observed pixels. In the Bayesian formulation, the data distribution $p_0$ serves as a prior over natural images, while the observation model is encoded by a Gaussian likelihood on the observed coordinates:

$$\ell_0(\mathbf{y}|\mathbf{x}) = \mathrm{N}\big(\mathbf{y}; \mathbf{x}[\overline{\mathbf{m}}], \sigma_{\mathbf{y}}^2 \mathrm{I}_{d_{\mathbf{y}}}\big) \,. \tag{2.4}$$

The parameter $\sigma_{\mathbf{y}} > 0$ serves as a relaxation factor: smaller values enforce strict adherence to the observation, while larger values permit controlled deviations from $\mathbf{x}_*$, thereby facilitating the reconstruction process. In this Bayesian framework, the target distribution from which we aim to sample is the *posterior distribution*

$$\pi_0(\mathbf{x}_0|\mathbf{y}) \propto \ell_0(\mathbf{y}|\mathbf{x}_0)\, p_0(\mathbf{x}_0) \,. \tag{2.5}$$

**Inference-time guidance.** As observed in the seminal works of Song & Ermon (2019); Kadkhodaie & Simoncelli (2020); Song et al. (2021b); Kawar et al. (2021), approximate sampling from the posterior distribution can be performed by biasing the denoising process with guidance terms, without requiring any additional fine-tuning. The central idea is to modify the sampling dynamics of diffusion

---

**Algorithm 1** Posterior sampling with decoupled guidance

---

1: **Input:** decreasing timesteps $(t_k)_{k=K}^0$ with $t_K = 1$, $t_0 = 0$; original image $\mathbf{x}_*$; mask $\mathbf{m}$;
   DDIM parameters $(\eta_k)_{k=K}^0$.
2: $\mathbf{y} \leftarrow \mathbf{x}_*[\overline{\mathbf{m}}]$; $\quad \mathbf{x} \sim \mathcal{N}(0, \mathrm{I}_d)$
3: **for** $k = K - 1$ **to** 1 **do**
4: $\quad \hat{\mathbf{x}}_0 \leftarrow \mathbf{x}_0^\theta(\mathbf{x}, t_{k+1})$
5: $\quad \hat{\mathbf{x}}_1 \leftarrow (\mathbf{x} - \alpha_{t_{k+1}} \hat{\mathbf{x}}_0) / \sigma_{t_{k+1}}$
6: $\quad \boldsymbol{\mu} \leftarrow \alpha_{t_k} \hat{\mathbf{x}}_0 + (\sigma_{t_k}^2 - \eta_k^2)^{1/2} \hat{\mathbf{x}}_1$
   /* Sampling (3.4) */
7: $\quad (\mathbf{w}, \mathbf{w}') \overset{\text{i.i.d.}}{\sim} \mathcal{N}(0_d, \mathrm{I}_d)$
8: $\quad \mathbf{z} \leftarrow \boldsymbol{\mu} + \eta_k \mathbf{w}$
9: $\quad \hat{\mathbf{x}}_1^{\text{pxy}} \leftarrow (\mathbf{z} - \alpha_{t_k} \hat{\mathbf{x}}_0^\theta(\mathbf{z}, t_k)) / \sigma_{t_k}$
10: $\quad \gamma \leftarrow \eta_{t_k}^2 / (\eta_{t_k}^2 + \alpha_{t_k}^2 \sigma_{\mathbf{y}}^2)$
11: $\quad \mathbf{x}[\mathbf{m}] \leftarrow \boldsymbol{\mu}[\mathbf{m}] + \eta_k \mathbf{w}'[\mathbf{m}]$
12: $\quad \mathbf{x}[\overline{\mathbf{m}}] \leftarrow (1 - \gamma) \boldsymbol{\mu}[\overline{\mathbf{m}}] + \gamma (\alpha_{t_k} \mathbf{y} + \sigma_{t_k} \hat{\mathbf{x}}_1^{\text{pxy}}[\overline{\mathbf{m}}]) + \alpha_{t_k} \sigma_{\mathbf{y}} \sqrt{\gamma} \mathbf{w}'[\overline{\mathbf{m}}]$
13: **end for**
14: **Return:** $\hat{\mathbf{x}}_0^\theta(\mathbf{x}, t_1)$

---

models on-the-fly so that the generated samples both satisfy the likelihood constraint $\ell_0(\mathbf{y}|\cdot)$ and remain plausible under the prior $p_0$. More precisely, a standard approach is to approximate the iterative updates of a diffusion model defined to target the posterior $\pi_0(\cdot|\mathbf{y})$. This in turn entails deriving an approximation of the posterior denoiser $\hat{\mathbf{x}}_0(\mathbf{x}_t, t|\mathbf{y}) := \int \mathbf{x}_0 \, \pi_{0|t}(\mathbf{x}_0|\mathbf{x}_t, \mathbf{y}) \, \mathrm{d}\mathbf{x}_0$, where $\pi_{0|t}(\mathbf{x}_0|\mathbf{x}_t, \mathbf{y}) \propto \pi_0(\mathbf{x}_0|\mathbf{y}) \mathrm{N}(\mathbf{x}_t; \alpha_t \mathbf{x}_0, \sigma_t^2 \mathrm{I})$. The denoiser $\hat{\mathbf{x}}_0(\cdot, t|\mathbf{y})$ is related to the prior denoiser via the identity

$$\hat{\mathbf{x}}_0(\mathbf{x}_t, t|\mathbf{y}) = \hat{\mathbf{x}}_0(\mathbf{x}_t, t) + \alpha_t^{-1} \sigma_t^2 \nabla_{\mathbf{x}_t} \log \ell_t(\mathbf{y}|\mathbf{x}_t) \,, \tag{2.6}$$

where the additional term is referred to as the *guidance term*; see Daras et al. (2024, Eq. 2.15 and 2.17). It is defined as the logarithmic gradient of the *propagated likelihood*

$$\ell_t(\mathbf{y}|\mathbf{x}_t) := \int \ell_0(\mathbf{y}|\mathbf{x}_0) \, p_{0|t}(\mathbf{x}_0|\mathbf{x}_t) \, \mathrm{d}\mathbf{x}_0 \,, \text{ with } p_{0|t}(\mathbf{x}_0|\mathbf{x}_t) \propto p_0(\mathbf{x}_0) \, \mathrm{N}(\mathbf{x}_t; \alpha_t \mathbf{x}_0, \sigma_t^2 \mathrm{I}) \,; \tag{2.7}$$

see Daras et al. (2024, Equation 2.20). Since the pre-trained parametric approximation $\hat{\mathbf{x}}_0^\theta(\cdot, t)$ of the prior denoiser $\hat{\mathbf{x}}_0(\cdot, t)$ is already available, estimating $\hat{\mathbf{x}}_0(\cdot, t|\mathbf{y})$ reduces to computing the intractable score term $\nabla_{\mathbf{x}_t} \log \ell_t(\mathbf{y}|\mathbf{x}_t)$. A widely adopted approximation (Ho et al., 2022; Chung et al., 2023) replaces $p_{0|t}(\cdot|\mathbf{x}_t)$ in (2.7) by a Dirac mass at the denoiser estimate $\hat{\mathbf{x}}_0^\theta(\mathbf{x}_t, t)$, yielding

$$\hat{\ell}_t^\theta(\mathbf{y}|\mathbf{x}_t) := \ell_0(\mathbf{y}|\hat{\mathbf{x}}_0^\theta(\mathbf{x}_t, t)) \,. \tag{2.8}$$

This approximation is often combined with a suitable rescaling weight (possibly depending on $\mathbf{x}_t$); see Ho et al. (2022, Equation 8) and Chung et al. (2023, Algorithm 1). Substituting this into the identity (2.6) yields an approximation of the posterior denoiser, which in turn defines an approximate diffusion model for $\pi_0(\cdot|\mathbf{y})$.

## 3 METHOD

The methods discussed in the previous section rely on the likelihood approximation (2.8), which is then inserted into (2.6). However, computing this term requires differentiating through the denoisers $\hat{\mathbf{x}}_0^\theta(\cdot, t_k)$ at each timestep $t_k$. This operation is computationally demanding: it increases memory usage, slows down the sampling process, and reduces scalability. By contrast, fine-tuned conditional diffusion models bypass these inference costs once training is complete, but at the expense of per-task retraining. This highlights a fundamental trade-off: zero-shot posterior sampling eliminates the need for retraining, but incurs substantial overhead during inference. Our goal is to bridge this gap by designing a zero-shot posterior sampler that removes the need for backpropagation through the denoiser while preserving the effectiveness of guidance.

**Reverse transitions for the posterior.** Our method builds upon the alternative sampling strategy introduced in Wu et al. (2023); Zhang et al. (2023); Janati et al. (2024). Instead of initializing the interpolation (2.1) with the prior $X_0 \sim p_0$, we consider the same process initialized from the posterior distribution $X_0 \sim \pi_0(\cdot|\mathbf{y})$. This yields a new family of random variables whose marginals are $\pi_t(\mathbf{x}_t|\mathbf{y}) \coloneqq \int \mathrm{N}(\mathbf{x}_t; \alpha_t \mathbf{x}_0, \sigma_t^2 \mathrm{I}_d) \pi_0(\mathbf{x}_0|\mathbf{y}) \, \mathrm{d}\mathbf{x}_0$, in analogy with the prior family $(p_t)_{t \in [0,1]}$. Moreover, the DDIM transitions associated with $(\pi_t(\cdot|\mathbf{y}))_{t \in [0,1]}$ are given by Janati et al. (2025b, Equation 1.17):

$$\pi_{s|t}^{\eta}(\mathbf{x}_s|\mathbf{x}_t, \mathbf{y}) \propto \ell_s(\mathbf{y}|\mathbf{x}_s) \, p_{s|t}^{\eta}(\mathbf{x}_s|\mathbf{x}_t) \,, \tag{3.1}$$

which defines a valid Markov chain with marginals $(\pi_{t_k}(\cdot|\mathbf{y}))_{k=1}^K$. This chain defines a path between the Gaussian reference $\mathcal{N}(0, \mathrm{I}_d)$ and the posterior distribution $\pi_0(\cdot|\mathbf{y})$. However, the presence of the likelihood term $\ell_t(\mathbf{y}|\mathbf{x}_t)$ makes also these transitions intractable. To address this issue, prior works (Zhang et al., 2023; Wu et al., 2023) introduced the surrogate transitions proportional to $\mathbf{x}_s \mapsto \hat{\ell}_s^{\theta}(\mathbf{y}|\mathbf{x}_s) p_{s|t}^{\eta,\theta}(\mathbf{x}_s|\mathbf{x}_t)$, for fixed $\mathbf{x}_t$ and $\mathbf{y}$, where $\hat{\ell}_t^{\theta}(\mathbf{y}|\cdot)$ are defined in (2.8). These transitions are then approximated using either variational inference (Janati et al., 2024; Pandey et al., 2025) or sequential Monte Carlo methods (Wu et al., 2023). However, similar to the methods described in the previous section, these approximations rely on the approximate guidance term and thus suffer from inflated memory usage and higher runtime.

**Our likelihood approximation.** To address this limitation, we draw inspiration from (2.8) to propose a lightweight approximation, designed to eliminate the need for VJP evaluations through the denoiser. Using the relation $\hat{\mathbf{x}}_0^{\theta}(\mathbf{x}_s, s) = (\mathbf{x}_s - \sigma_s \hat{\mathbf{x}}_1^{\theta}(\mathbf{x}_s, s))/\alpha_s$, we first rewrite the standard likelihood approximation (2.8) in terms of the noise prediction $\hat{\mathbf{x}}_1^{\theta}(\mathbf{x}_s, s)$ according to

$$\hat{\ell}_s^{\theta}(\mathbf{y}|\mathbf{x}_s) = \ell_0(\mathbf{y}|(\mathbf{x}_s - \sigma_s \hat{\mathbf{x}}_1^{\theta}(\mathbf{x}_s, s))/\alpha_s) \,.$$

Based on this parametrization, we then introduce the following alternative approximation

$$\hat{\ell}_s^{\theta}(\mathbf{y}|\mathbf{x}_s, \mathbf{z}_s) \coloneqq \ell_0(\mathbf{y}|(\mathbf{x}_s - \sigma_s \hat{\mathbf{x}}_1^{\theta}(\mathbf{z}_s, s))/\alpha_s) \,, \tag{3.2}$$

where the noise predictor is evaluated at $\mathbf{z}_s \in \mathbb{R}^d$, which serves as a proxy for $\mathbf{x}_s$. A key feature of this decoupling is that it enables lightweight updates, avoids costly denoiser backpropagation, and still provides high-quality reconstructions. Then, similarly to (3.1), we define

$$\hat{\pi}_{s|t}^{\theta}(\mathbf{x}_s|\mathbf{z}_s, \mathbf{x}_t, \mathbf{y}) \propto \hat{\ell}_s^{\theta}(\mathbf{y}|\mathbf{x}_s, \mathbf{z}_s) \, p_{s|t}^{\eta,\theta}(\mathbf{x}_s|\mathbf{x}_t) \,.$$

This leads us to propose the surrogate[1]

$$\hat{\pi}_{s|t}^{\theta}(\mathbf{x}_s|\mathbf{x}_t, \mathbf{y}) \coloneqq \mathbb{E}\left[\hat{\pi}_{s|t}^{\theta}(\mathbf{x}_s|Z_s, \mathbf{x}_t, \mathbf{y})\right] \,, \tag{3.3}$$

where $Z_s \sim p_{s|t}^{\theta}(\cdot|\mathbf{x}_t)$, for (3.1). The transition $\hat{\pi}_{s|t}^{\theta}(\mathbf{x}_s|\mathbf{x}_t, \mathbf{y})$ generally lacks a closed-form expression; nevertheless, since it has a *mixture structure*, it allows for straightforward and efficient sampling. Sampling from $\hat{\pi}_{s|t}^{\theta}(\mathbf{x}_s|\mathbf{x}_t, \mathbf{y})$ can be performed by first drawing $Z_s$ from $p_{s|t}^{\theta}(\cdot|\mathbf{x}_t)$, and then sampling from $\hat{\pi}_{s|t}^{\theta}(\mathbf{x}_s|Z_s, \mathbf{x}_t, \mathbf{y})$. Moreover, as we will now show, in the case of inpainting, the second step can be carried out exactly.

Let $\boldsymbol{\mu}_{s|t}^{\theta}(\mathbf{x}_t; \eta)$ denote the mean of the Gaussian reverse transition $p_{s|t}^{\eta,\theta}(\cdot|\mathbf{x}_t)$. In the case of inpainting (2.4), standard Gaussian conjugacy results (Bishop, 2006, Equation 2.116) show that $\hat{\pi}_{s|t}^{\theta}(\cdot|\mathbf{z}_s, \mathbf{x}_t, \mathbf{y})$ admits a closed-form Gaussian expression

$$\hat{\pi}_{s|t}^{\theta}(\mathbf{x}_s|\mathbf{z}_s, \mathbf{x}_t, \mathbf{y}) = \mathrm{N}\big(\mathbf{x}_s[\mathbf{m}]; \boldsymbol{\mu}_{s|t}^{\theta}(\mathbf{x}_t; \eta)[\mathbf{m}], \eta_s^2 \mathrm{I}_{d-d_\mathbf{y}}\big)$$
$$\times \mathrm{N}\big(\mathbf{x}_s[\overline{\mathbf{m}}]; (1 - \gamma_{s|t})\boldsymbol{\mu}_{s|t}^{\theta}(\mathbf{x}_t; \eta)[\overline{\mathbf{m}}] + \gamma_{s|t}(\alpha_s \mathbf{y} + \sigma_s \hat{\mathbf{x}}_1^{\theta}(\mathbf{z}_s, s)[\overline{\mathbf{m}}]), \alpha_s^2 \sigma_\mathbf{y}^2 \gamma_{s|t} \mathrm{I}_{d_\mathbf{y}}\big) \,, \tag{3.4}$$

with $\gamma_{s|t} \coloneqq \eta_s^2/(\eta_s^2 + \alpha_s^2 \sigma_\mathbf{y}^2)$. A derivation is provided in Appendix A.1. Thus, a sample $X_s$ from (3.3) can be drawn exactly by, first, generating a realization $\mathbf{z}_s$ of $Z_s \sim p_{s|t}^{\eta,\theta}(\cdot|\mathbf{x}_t)$ and, second, sampling $X_s[\mathbf{m}]$ and $X_s[\overline{\mathbf{m}}]$ conditionally independently from the two Gaussian distributions in (3.4); see Algorithm 1 for a pseudocode of this approach, which we refer to as DING (see Section 1).

---

[1]In the follow-up work Ghorbel et al. (2026), we provide further insight into this approximation.

**Practical implementation.** A key practical feature of our method is that it depends on a single hyperparameter: the sequence $(\eta_t)_{t \in [0,1]}$ of standard deviations, which controls the level of stochasticity in the DDIM reverse process. This choice is particularly critical in the low-NFE regime, where only a few function evaluations are available and the variance schedule strongly influences both observation fidelity and perceptual quality. In all experiments, we adopt the schedule $\eta_t = \sigma_t(1 - \alpha_t)$. An ablation study of this parameter is reported in Section 4.

Beyond this hyperparameter, an important practical consideration is that most large-scale diffusion models for high-resolution image generation operate in a compressed latent space rather than in pixel space (Rombach et al., 2022; Esser et al., 2024). To apply our algorithm in this setting, we must therefore formulate the inpainting task in the latent space. Denote by Enc the encoder, $\mathbf{X}_*$ the pixel-space ground-truth image and $\mathbf{M}$ the corresponding pixel-space mask. Following Avrahami et al. (2022), we set $\mathbf{x}_* := \mathsf{Enc}(\mathbf{X}_*)$, the observation to $\mathbf{y} := \mathbf{x}_*[\mathbf{m}]$ where $\mathbf{m}$ is a downsampled version of the pixel-space mask $\mathbf{M}$. We illustrate in Figure 3 that masking in the latent spaces translates to masking in the pixel despite the nonlinearity of the decoder. Since the encoder reduces spatial resolution by a fixed factor (*e.g.*, $s = 8$ in Esser et al. (2024)), we construct the latent mask $\mathbf{m}$ by average pooling the binary pixel-space mask $\mathbf{M}$ with kernel and stride $s$. Each latent site is assigned the fraction of unmasked pixels within its receptive field. These fractional values are then thresholded (typically at $0.5$) to produce a binary mask; in other words, a latent site is marked as observed if the majority of its underlying pixels are unmasked. In practice, the mask $\mathbf{m}$ is provided as a single-channel image and broadcast across all latent channels when applied to $\mathbf{x}_*$. Finally, we apply Algorithm 1 with $(\mathbf{x}_*, \mathbf{y}, \mathbf{m})$ thus defined in the latent space.

**Related methods.** Our work shares similarities with various recent approaches to zero-shot diffusion guidance, which now briefly review. The closest line of work comprises variants of the *replacement method* (Song & Ermon, 2019; Song et al., 2021b), which follows the same structure as Algorithm 1. In these schemes, the masked coordinates of the state are updated according to the standard DDIM transition (Line 11), while the unmasked coordinates are replaced by a direct update that enforces consistency with the observation $\mathbf{y}$ (Line 12). In its simplest form, the method performs ancestral sampling with the transition

$$\hat{\pi}^\theta_{s|t}(\mathbf{x}_s|\mathbf{x}_t, \mathbf{y}) = \mathrm{N}\big(\mathbf{x}_s[\mathbf{m}]; \, \boldsymbol{\mu}^\theta_{s|t}(\mathbf{x}_t;\eta)[\mathbf{m}], \, \eta_s^2 \mathrm{I}_{d-d_{\mathbf{y}}}\big) \, \mathrm{N}\big(\mathbf{x}_s[\overline{\mathbf{m}}]; \, \alpha_s \mathbf{y}, \, \sigma_s^2 \mathrm{I}_{d_{\mathbf{y}}}\big) \,, \qquad (3.5)$$

*i.e.*, the unmasked state is set to a noisy version of the observation $\alpha_s \mathbf{y} + \sigma_s W_s$, where $W_s \sim \mathcal{N}(0, \mathrm{I}_{d_{\mathbf{y}}})$; see Song & Ermon (2019, Algorithm 2) and Song et al. (2021b, Appendix I.2). Avrahami et al. (2022) extended this approach to the latent space using a downsampled mask. The method was later refined in RePaint (Lugmayr et al., 2022), which improves sample quality by performing multiple back-and-forth updates: after applying the replacement step from $t_{k+1}$ to $t_k$, a forward noising step is applied from $t_k$ back to $t_{k+1}$, and this cycle is repeated several times. Several works have combined the replacement method with sequential Monte Carlo (SMC) sampling (Trippe et al., 2023; Cardoso et al., 2023; Dou & Song, 2024; Corenflos et al., 2025; Zhao, 2025). In particular, Cardoso et al. (2023) update the unmasked coordinates of each particle using a Gaussian transition whose mean is a convex combination of the DDIM mean and the rescaled observation $\alpha_{t_k} \mathbf{y}$. In the inpainting framework, the recently proposed PnP-Flow (Martin et al., 2025) reduces to using similar transitions without relying on SMC, *i.e.*, by using a single particle. We explicitly compare the update rules in Cardoso et al. (2023); Martin et al. (2025) with ours in Appendix A.2, where we also discuss additional related work.

## 4 EXPERIMENTS

In this section, we extensively evaluate the inpainting performance of DInG when used with different large-scale models. We benchmark its performance on multiple datasets against several state-of-the-art baselines. We further analyze the relevance of our modeling choices, specifically the formulation of the approximation in (3.2) and the schedule of DDIM standard deviations $(\eta_t)_{t \in [0,1]}$, through a series of targeted ablations.

**Models and datasets.** We evaluate our method on Stable Diffusion 3.5 (medium) (Esser et al., 2024). We set the CFG scale to 2. Our experiments cover three datasets: FFHQ (Karras et al., 2019), DIV2K (Agustsson & Timofte, 2017), and PIE-Bench (Ju et al., 2024). For FFHQ, we use the first

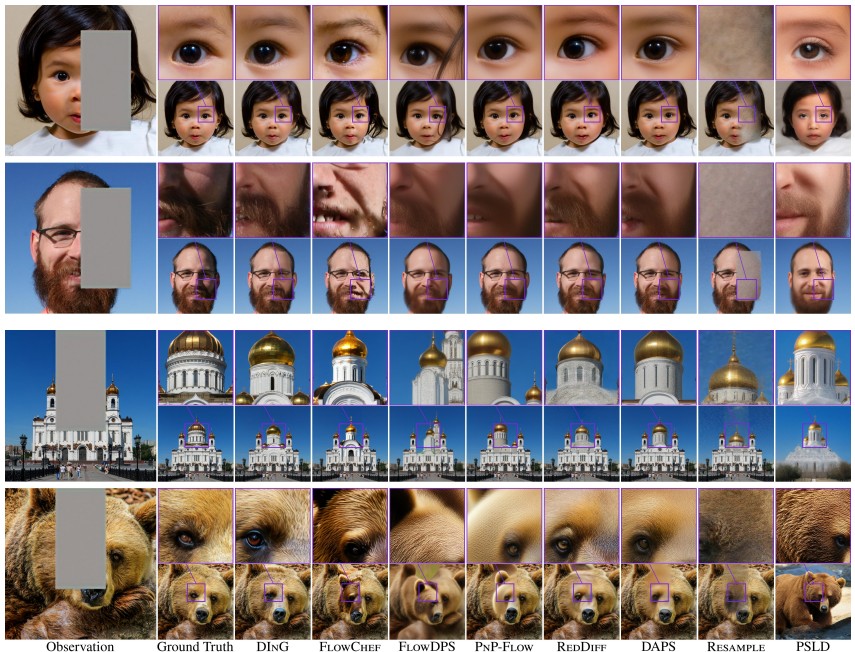

Figure 2: Examples of reconstructions on `FFHQ` and `DIV2K` with 50 NFEs.

5k images and condition generation on the prompt *"a high-quality photo of a face"*. For `DIV2K`, we include both training and validation splits (900 images in total), and generate captions for each image using BLIP-2 (Li et al., 2023); see Appendix B for details. All `FFHQ` and `DIV2K` images are resized to a resolution of $768 \times 768$. The `PIE-Bench` dataset contains 700 images of resolution $512 \times 512$, each paired with an inpainting mask and an edit caption. After removing cases where the mask completely covers the image, the resulting evaluation set contains 556 images.

**Evaluation and masks.** For `FFHQ` and `DIV2K`, we evaluate inpainting performance under four rectangular masking configurations: (i) right half of the image (*Half*), (ii) upper half (*Top*), (iii) lower half (*Bottom*), and (iv) a central $512 \times 512$ square (*Center*). In contrast, `PIE-Bench` provides irregular masks with diverse spatial patterns; see Appendix C for examples. Unless otherwise stated, we set $\sigma_\mathbf{y} = 0.01$ across all tasks. Since exact posterior sampling is infeasible, we assess inpainting quality using proxy metrics. To measure distributional alignment with the dataset, we report both FID and patch FID (pFID) (Chai et al., 2022), the latter offering finer granularity for high-resolution evaluation. Following the standard FID protocol, we extract 10 random $256 \times 256$ patches per image, yielding a total of 50k patches. To quantify consistency with the observed content, we compute context PSNR (cPSNR), defined as the PSNR over the unmasked region only. We further report LPIPS (Zhang et al., 2018) relative to the ground truth to evaluate perceptual similarity, which is especially relevant for `FFHQ` where facial symmetries make reconstructions visually close to the reference. For `PIE-Bench`, which includes edit captions, we additionally report CLIP-Score (Radford et al., 2021) on both the full image (CLIP) and the edited region (CLIP-ED), following Ju et al. (2024). Together, these metrics provide a comprehensive evaluation of inpainting quality. While each captures a different aspect of performance, none should be interpreted in isolation.

**Baselines.** We compare against seven state-of-the-art baselines: FLOWCHEF (Patel et al., 2024), FLOWDPS (Kim et al., 2025), DAPS (Zhang et al., 2025), REDDIFF (Mardani et al., 2024), RESAMPLE (Song et al., 2024), PSLD (Rout et al., 2024b), PNP-FLOW (Martin et al., 2025), DIFFPIR (Zhu et al., 2023), DDNM (Wang et al., 2023b) and BLENDED-DIFF (Avrahami et al., 2023) For the main comparison, all methods are evaluated under a fixed budget of 50 NFEs. Since our method requires two denoiser evaluations per diffusion step, we use 25 steps to match this budget. We focus on this low-NFE regime as it reflects realistic settings, where inference is constrained by latency and compute. To ensure

Table 1: Memory and runtime.

| Method | DIV2K 768px | |
| | Time (s) | Mem. (GB) |
| --- | --- | --- |
| BLENDED-DIFF | 3.0 | 22.09 |
| DAPS | 9.1 | 22.09 |
| DDNM | 3.1 | 22.09 |
| DIFFPIR | 3.1 | 22.09 |
| FLOWCHEF | 3.0 | 22.09 |
| FLOWDPS | 3.0 | 22.10 |
| PNP-FLOW | 3.1 | 22.09 |
| PSLD | 7.4 | 24.49 |
| REDDIFF | 3.1 | 22.09 |
| RESAMPLE | 8.1 | 24.50 |
| DING (ours) | 2.9 | 22.09 |

Table 2: **Top**: Quantitative results on FFHQ 768 × 768 with 5k samples. **Bottom**: DIV2K 768 × 768 with 900 samples. For FID, pFID, and LPIPS, the lower the better. For cPSNR, the higher the better. 50 NFEs were used.

| | Half | | | | Center | | | | Top | | | | Bottom | | | |
|---|---|---|---|---|---|---|---|---|---|---|---|---|---|---|---|---|
| Method | FID | pFID | cPSNR | LPIPS | FID | pFID | cPSNR | LPIPS | FID | pFID | cPSNR | LPIPS | FID | pFID | cPSNR | LPIPS |
| | | | | | | | | FFHQ 768 × 768 | | | | | | | | |
| BLENDED-DIFF | 23.5 | 16.3 | **31.32** | 0.38 | 35.3 | 36.7 | 31.54 | 0.33 | 32.8 | 15.8 | 32.05 | 0.38 | 43.7 | 19.8 | **30.85** | 0.37 |
| DAPS | 17.9 | 25.1 | 30.50 | 0.36 | 35.1 | 54.5 | 31.15 | 0.32 | 30.1 | 30.5 | 31.54 | 0.39 | 52.8 | 27.6 | 30.30 | 0.34 |
| DDNM | 12.3 | 13.68 | 31.27 | **0.33** | 24.4 | 34.8 | 31.61 | 0.27 | 22.3 | 23.2 | 31.82 | 0.36 | 38.3 | 19.6 | 30.51 | 0.32 |
| DIFFPIR | 12.1 | 11.23 | 30.91 | 0.36 | 19.4 | 19.6 | **31.67** | 0.30 | 19.7 | 14.1 | 32.07 | 0.36 | 30.7 | 11.4 | 30.74 | 0.35 |
| FLOWCHEF | 20.2 | 16.5 | 30.41 | 0.36 | 29.3 | 35.0 | 31.00 | 0.31 | 27.8 | 21.1 | 31.05 | 0.36 | 35.9 | 22.7 | 29.94 | 0.35 |
| FLOWDPS | 36.2 | 49.0 | 26.72 | 0.46 | 49.5 | 79.9 | 23.36 | 0.53 | 52.1 | 74.2 | 24.15 | 0.56 | 72.3 | 71.5 | 23.06 | 0.55 |
| PNP-FLOW | 20.5 | 33.4 | 30.62 | 0.37 | 36.6 | 65.1 | **31.67** | 0.32 | 33.6 | 42.7 | 31.54 | 0.38 | 56.8 | 33.4 | 29.95 | 0.33 |
| PSLD | 116.3 | 73.8 | 6.89 | 0.81 | 98.0 | 69.1 | 10.09 | 0.73 | 120.6 | 75.4 | 7.06 | 0.81 | 107.0 | 70.4 | 6.46 | 0.81 |
| REDDIFF | 28.5 | 37.9 | 27.39 | 0.39 | 30.7 | 41.8 | 27.85 | 0.32 | 33.0 | 41.1 | 27.92 | 0.41 | 76.4 | 41.3 | 26.96 | 0.39 |
| RESAMPLE | 32.4 | 48.8 | 28.53 | 0.44 | 53.8 | 103.4 | 28.46 | 0.40 | 63.2 | 56.2 | 29.02 | 0.44 | 97.8 | 57.0 | 28.06 | 0.44 |
| DING (ours) | **9.6** | **6.6** | 31.03 | **0.33** | **15.5** | **14.0** | 31.38 | **0.27** | **19.7** | **12.5** | 31.64 | **0.34** | **29.6** | **8.6** | 30.50 | **0.32** |
| | | | | | | | | DIV2K 768 × 768 | | | | | | | | |
| BLENDED-DIFF | 43.6 | 12.9 | 26.03 | 0.37 | 54.8 | 20.2 | 26.43 | 0.35 | 44.8 | 13.2 | 25.28 | 0.39 | 48.1 | 13.1 | 26.85 | 0.38 |
| DAPS | 51.0 | 38.4 | 25.92 | 0.46 | 74.8 | 67.6 | 26.14 | 0.44 | 54.8 | 41.0 | 25.22 | 0.44 | 61.2 | 39.7 | 26.71 | 0.50 |
| DDNM | 42.5 | 21.2 | 26.03 | 0.41 | 57.7 | 38.5 | 26.61 | 0.37 | 45.7 | 21.3 | 25.36 | 0.42 | 49.6 | 23.2 | 26.81 | 0.45 |
| DIFFPIR | 41.1 | 12.9 | 26.09 | 0.37 | 52.8 | 21.4 | 26.58 | 0.34 | 43.5 | 13.1 | 25.36 | 0.39 | 44.9 | 14.9 | 26.91 | 0.39 |
| FLOWCHEF | 43.3 | 12.2 | 25.78 | 0.36 | 53.6 | 22.3 | 26.27 | 0.32 | 45.0 | 13.8 | 25.09 | 0.37 | 46.9 | 13.2 | 26.57 | 0.37 |
| FLOWDPS | 50.8 | 33.2 | 21.30 | 0.49 | 70.3 | 62.8 | 18.38 | 0.63 | 64.1 | 57.9 | 17.43 | 0.65 | 64.2 | 57.3 | 19.06 | 0.63 |
| PNP-FLOW | 54.2 | 42.7 | 26.00 | 0.46 | 79.7 | 71.4 | **26.63** | 0.44 | 57.1 | 33.5 | 25.19 | 0.44 | 64.7 | 50.8 | 26.61 | 0.50 |
| PSLD | 66.4 | 32.3 | 6.15 | 0.79 | 66.9 | 35.7 | 9.89 | 0.72 | 66.5 | 31.9 | 6.35 | 0.79 | 66.3 | 32.6 | 6.41 | 0.78 |
| REDDIFF | 54.2 | 45.7 | 22.64 | 0.49 | 77.4 | 69.8 | 23.25 | 0.46 | 57.7 | 40.9 | 22.17 | 0.48 | 60.6 | 46.3 | 23.41 | 0.52 |
| RESAMPLE | 52.7 | 34.1 | 23.33 | 0.47 | 80.8 | 63.8 | 23.76 | 0.43 | 56.1 | 33.1 | 22.84 | 0.47 | 60.6 | 41.0 | 24.06 | 0.48 |
| DING (ours) | **39.2** | 13.0 | 25.90 | **0.35** | **50.7** | 19.5 | 26.41 | **0.31** | **41.4** | 13.7 | 25.19 | **0.37** | **43.4** | 13.4 | 26.72 | **0.37** |

fairness, all methods are run in the latent space with downsampled masks, and extensive hyperparameter tuning is performed for each baseline on each dataset. For baselines that require VJP or backpropagation through the denoiser, we report their actual runtime and memory costs, ensuring that comparisons reflect effective inference cost rather than nominal NFE counts. Average runtime and memory usage across all the experiments, measured on H100 GPUs, are provided in Table 1.

## 4.1 MAIN RESULTS

Tables 2 and 3 summarize the results on FFHQ, DIV2K and PIE-Bench, respectively. On FFHQ (Table 3), DING achieves the best performance on all masks and almost all the metrics. In particular, it improves both pFID and FID by significant margins over the strongest competing method FLOWCHEF. It also obtains the highest cPSNR scores, indicating a faithful reconstruction of the visible content, while simultaneously producing visually coherent completions with the lowest LPIPS. On DIV2K, the comparison is more nuanced. DING consistently attains the best FID and LPIPS across all four masks, while remaining competitive on pFID and comparable to strong

Table 3: Results on PIE-Bench with 556 samples and 50 NFEs.

| Method | FID | pFID | cPSNR | LPIPS | CLIP | CLIP-ED |
|---|---|---|---|---|---|---|
| BLENDED-DIFF | 65.5 | 27.0 | 26.60 | 0.31 | 26.32 | 23.15 |
| DAPS | 65.9 | 30.2 | 27.08 | 0.34 | 25.57 | 21.75 |
| DDNM | **61.4** | 26.9 | **27.29** | 0.31 | 26.27 | 22.96 |
| DIFFPIR | 63.5 | 25.4 | 26.98 | **0.30** | 26.21 | 23.04 |
| FLOWCHEF | 68.3 | 27.4 | 26.84 | **0.30** | 26.02 | 22.47 |
| FLOWDPS | 74.6 | 42.7 | 22.05 | 0.45 | **26.35** | 22.79 |
| PNP-FLOW | 66.8 | 32.1 | 26.90 | 0.34 | 25.62 | 21.02 |
| PSLD | 94.1 | 56.2 | 14.25 | 0.65 | 26.32 | 21.81 |
| REDDIFF | 69.5 | 35.2 | 24.34 | 0.37 | 25.27 | 21.18 |
| RESAMPLE | 71.0 | 33.9 | 24.45 | 0.35 | 25.71 | 22.03 |
| DING (ours) | **61.4** | **24.7** | 27.03 | **0.30** | 26.30 | **23.36** |

baselines on cPSNR. On PIE-Bench, DING achieves the best results on all metrics except cPSNR and CLIP. We note, however, that although FLOWDPS and PSLD obtain slightly higher CLIP scores, they perform markedly worse on fidelity and perceptual quality metrics, suggesting that their improvements in CLIP may reflect metric hacking rather than genuine reconstruction quality. We provide qualitative comparisons of the reconstruction in Figure 7 and Appendix C.

We now compare DING with a Stable Diffusion 3 model fine-tuned for inpainting[2], trained on 12M images at 1024 × 1024 resolution. To ensure fairness, both models are evaluated under the same runtime budget (**2.2s**), which corresponds to 56 NFEs for DING and 28 NFEs for the fine-tuned baseline. We also provide the results for the finetuned model using 56 NFEs.

The results are given in Tables 5 and 4. Across FFHQ, DIV2K, and PIE-Bench, DING consistently outperforms the fine-tuned SD3 model on all reported metrics. The gains are especially pronounced in cPSNR, where DING achieves 8–10 dB higher fidelity to the observed pixels. This indicates that our method preserves the known content far more accurately while still producing realistic completions, as confirmed by

Table 4: Results on the PIE-Bench with 556 samples.

| | FID | pFID | cPSNR | LPIPS | CLIP | CLIP-ED |
|---|---|---|---|---|---|---|
| 3 seconds | | | | | | |
| SD3 Inpaint (28) | 68.7 | 30.5 | 18.85 | 0.34 | 26.37 | 23.10 |
| DING (ours) | **63.6** | **24.6** | **26.98** | **0.30** | **26.63** | **23.70** |

[2]https://huggingface.co/alimama-creative/SD3-Controlnet-Inpainting

lower FID and LPIPS. On `PIE-Bench`, DING further improves over the fine-tuned baseline on every metric, including perceptual ones (pFID, LPIPS), while also yielding stronger text–image alignment (CLIP, CLIP-ED). See Figure 7 for a qualitative comparison of the reconstructions. These results demonstrate that, even without task-specific fine-tuning on a large amount of images, DING not only matches but surpasses a specialized SD3 inpaint model. Overall, these results show that our method provides the strongest overall trade-off between realism and fidelity under low NFE budgets.

Table 5: DING compared to SD3 fine-tuned (28 and 56 NFEs) for inpainting tasks.

| Method | Half | | | | Center | | | | Top | | | | Bottom | | | |
|---|---|---|---|---|---|---|---|---|---|---|---|---|---|---|---|---|
| | FID | pFID | cPSNR | LPIPS | FID | pFID | cPSNR | LPIPS | FID | pFID | cPSNR | LPIPS | FID | pFID | cPSNR | LPIPS |
| | **FFHQ** $512 \times 512$ | | | | | | | | | | | | | | | |
| SD3 Inpaint (28) | 23.5 | 10.7 | 21.69 | 0.37 | 62.1 | 33.9 | 22.18 | 0.31 | 34.7 | 17.8 | 21.64 | 0.36 | 42.4 | 16.5 | 21.78 | 0.37 |
| SD3 Inpaint (56) | 23.7 | 10.3 | 21.53 | 0.37 | 63.7 | 34.4 | 21.94 | 0.31 | 35.4 | 16.5 | 21.41 | 0.36 | 43.8 | 16.8 | 21.53 | 0.36 |
| DING (ours) | **9.3** | **5.8** | **31.40** | **0.32** | **20.2** | **15.5** | **31.39** | **0.28** | **17.3** | **8.4** | **31.96** | **0.33** | **33.8** | **12.2** | **31.27** | **0.34** |
| | **DIV2K** $512 \times 512$ | | | | | | | | | | | | | | | |
| SD3 Inpaint (28) | 45.9 | 15.0 | 17.95 | 0.40 | 54.2 | 22.1 | 18.57 | 0.36 | 48.8 | 16.6 | 18.12 | 0.42 | 51.0 | 17.5 | 18.95 | 0.41 |
| SD3 Inpaint (56) | 45.1 | **14.0** | 17.91 | 0.40 | 54.2 | 20.5 | 18.63 | 0.36 | 48.6 | 16.1 | 18.16 | 0.41 | 50.3 | 17.2 | 19.02 | 0.41 |
| DING (ours) | **41.5** | 14.2 | **26.09** | **0.37** | **52.4** | **21.5** | **26.53** | **0.33** | **43.7** | **13.8** | **25.47** | **0.38** | **45.4** | **15.4** | **26.94** | **0.38** |

## 4.2 ABLATIONS

**Doubled NFE per diffusion step.** Because the $\mathbf{x}_1$-predictor must be evaluated at the proxy variable (Line 9 in Algorithm 1), our algorithm requires two NFEs per diffusion step. An immediate question is whether this overhead is needed. To explore this, we introduce a variant in which the noise prediction from the previous step is reused instead of being recomputed at the proxy. We coin this variant as Delayed DING, where Line 9 is replaced by $\hat{\mathbf{x}}_1^{\text{pxy}} \leftarrow (\mathbf{x} - \sigma_{t_k}\hat{\mathbf{x}}_1(\mathbf{x}, t_{k+1}))/\alpha_{t_k}$, and we further set $\eta_t = \sigma_t(1 - \alpha_t)$, which we found to yield the best performance in this setting. A quantitative comparison with the original DING is reported in Table 6, showing that while Delayed DING reduces the NFE cost per step, it consistently underperforms across metrics and masking patterns, indicating that the doubled NFE is necessary to retain the full effectiveness of our approach.

Table 6: Delayed DING compared to DING on FFHQ (5k samples) and DIV2K (900 samples) with 50 NFEs.

| Method | Half | | | | Center | | | | Top | | | | Bottom | | | |
|---|---|---|---|---|---|---|---|---|---|---|---|---|---|---|---|---|
| | pFID | FID | cPSNR | LPIPS | pFID | FID | cPSNR | LPIPS | pFID | FID | cPSNR | LPIPS | pFID | FID | cPSNR | LPIPS |
| | **FFHQ** $768 \times 768$ | | | | | | | | | | | | | | | |
| Delayed DING | **9.1** | 7.4 | 29.21 | 0.33 | 21.3 | 20.7 | 29.90 | **0.26** | **15.7** | 9.9 | 29.84 | 0.33 | 31.0 | 12.3 | 28.88 | 0.33 |
| DING | 9.6 | **6.6** | **31.03** | 0.33 | **15.5** | **14.0** | **31.38** | 0.27 | 19.7 | **12.5** | **31.64** | 0.34 | **29.6** | **8.6** | **30.50** | **0.32** |
| | **DIV2K** $768 \times 768$ | | | | | | | | | | | | | | | |
| Delayed DING | 43.9 | 15.9 | 24.88 | 0.36 | 55.6 | 24.7 | 25.52 | 0.32 | 45.3 | 16.8 | 24.36 | 0.38 | 47.8 | 14.6 | 25.64 | 0.38 |
| DING | **39.2** | **13.0** | **25.90** | **0.35** | **50.7** | **19.5** | **26.41** | **0.31** | **41.4** | **13.7** | **25.19** | **0.37** | **43.4** | **13.4** | **26.72** | **0.37** |

**DDIM schedule.** Here we proceed to compare the behavior of our algorithm under different schedules $(\eta_t)_{t \in [0,1]}$. For this purpose we compare against some natural candidates. (A): we consider the DDPM schedule used in Ho et al. (2020) and which corresponds to using in (2.2) a standard deviation that depends on both $s$ and $t$, *i.e.*, $\eta_s(t) = \sigma_s(\sigma_t^2 - (\alpha_t/\alpha_s)^2\sigma_s^2)^{1/2}/\sigma_t$. (B): as we cannot use deterministic sampling in our approach, we rescale the DDPM schedule (A) with $0.01$ to approach deterministic sampling. (C): $\eta_s = \sigma_s$, which is the maximum allowed standard deviation in (2.2). In this scenario, the transition is $p_{s|t}^\theta(\mathbf{x}_s|\mathbf{x}_t) = \text{N}(\mathbf{x}_s; \hat{\mathbf{x}}_0^\theta(\mathbf{x}_t, t), \sigma_s^2 \text{I}_d)$ and resembles the prior transition used in Martin et al. (2025). (D): $\eta_s = \sigma_s\sqrt{1 - \alpha_s}$, which corresponds to a slower decay of the standard deviation compared to our default choice $\eta_s = \sigma_s(1 - \alpha_s)$. The results on FFHQ with 5k samples and 50 NFEs are reported in Table 7. We observe that the rescaled DDPM schedule (B) degrades significantly across all metrics, while (A) and (C) yield nearly identical performance. This suggests that maintaining sufficient stochasticity at the beginning of the diffusion process is crucial for strong performance. Among the alternatives, (D) performs best; still, it is outperformed by our default schedule, confirming the benefit of a faster decay of $(\eta_t)$.

Table 7: Ablation results for the DDIM schedule ($\eta_t$) on FFHQ $768 \times 768$ with 5k samples and 50 NFEs.

| Method | Half | | | | Center | | | | Top | | | | Bottom | | | |
|---|---|---|---|---|---|---|---|---|---|---|---|---|---|---|---|---|
| | FID | pFID | cPSNR | LPIPS | FID | pFID | cPSNR | LPIPS | FID | pFID | cPSNR | LPIPS | FID | pFID | cPSNR | LPIPS |
| (A) | 13.9 | 14.0 | 31.19 | 0.36 | 19.1 | 25.5 | 31.50 | 0.30 | 21.8 | 18.9 | 31.80 | 0.38 | 35.1 | 15.1 | 30.70 | 0.35 |
| (B) | 21.5 | 18.7 | 26.06 | 0.41 | 29.0 | 31.9 | 26.23 | 0.35 | 31.7 | 21.1 | 26.64 | 0.41 | 48.4 | 28.6 | 25.56 | 0.40 |
| (C) | 13.9 | 14.2 | 31.19 | 0.36 | 19.1 | 25.5 | 31.50 | 0.30 | 21.8 | 19.0 | 31.80 | 0.38 | 35.1 | 15.6 | 30.70 | 0.35 |
| (D) | 10.2 | 10.7 | **31.33** | 0.33 | 16.7 | 19.0 | **31.70** | 0.27 | **19.6** | 15.7 | **31.95** | 0.35 | 31.6 | 12.0 | **30.81** | 0.32 |
| Default | **9.6** | **6.6** | 31.03 | **0.33** | **15.5** | **14.0** | 31.38 | **0.27** | 19.7 | **12.5** | 31.64 | **0.34** | **29.6** | **8.6** | 30.50 | **0.32** |

## 5  CONCLUSION

We have introduced DING, a novel diffusion-based method for zero-shot inpainting that operates fully in the latent space and enables fast, memory-efficient inference under low-NFE budgets. Through extensive experiments across multiple benchmarks, we have shown that DING consistently outperforms existing zero-shot approaches and even surpasses a fine-tuned Stable Diffusion 3 model for image editing, despite requiring no expensive training. Notably, our method produces globally coherent reconstructions while preserving the visible content with high fidelity.

**Limitations and future directions.**   While these results highlight the effectiveness and practicality of DING, several avenues remain open. An important limitation of our current approach is that performance does not monotonically improve as the compute budget increases. Ideally, one would like reconstruction accuracy to keep improving with additional sampling steps, potentially beyond the standard diffusion horizon, but we observe diminishing returns due to the limitations of our current DDIM schedule. Addressing this issue, for example by designing guidance schemes or noise schedules that continue to scale gracefully with compute, remains an important direction for future work. Moreover, while our framework is fully operational in the latent space, its applicability is currently limited to inpainting, as this is the only observation operator we can reliably lift to the latent space. Extending the method to accommodate more general forward operators and a broader class of inverse problems, while preserving the same level of efficiency achieved for inpainting, is a challenging yet promising direction for future research.

**Reproducibility statement.**   We place strong emphasis on reproducibility. To this end, we provide the full source code of our method along with implementations of all baseline methods used in the paper. Our repository also includes scripts to reproduce every experiment, as well as configuration files specifying all hyperparameters and settings for each baseline and experimental setup. Together, these resources ensure that all results reported in this work can be fully reproduced and easily extended.

**Ethics statement.**   While the proposed approach demonstrates clear benefits for applications in restoration, accessibility, and creative media, it also lies at the borderline of ethical considerations. Diffusion-based inpainting methods can be misappropriated for producing deceptive or harmful content, such as manipulated images or synthetic media that obscure authenticity. This dual-use nature highlights the need for proactive safeguards, including transparent usage guidelines, traceable model outputs, and continued development of forensic detection tools to ensure responsible integration of such technologies.

## ACKNOWLEDGEMENTS

The work of Badr Moufad has been supported by Technology Innovation Institute (TII), project Fed2Learn. The work of Eric Moulines has been partly funded by the European Union (ERC-2022-SYG-OCEAN-101071601). Views and opinions expressed are however those of the author(s) only and do not necessarily reflect those of the European Union or the European Research Council Executive Agency. Neither the European Union nor the granting authority can be held responsible for them. This work was granted access to the HPC resources of IDRIS under the allocations 2025-AD011015980 and 2025-AD011016484 made by GENCI.

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

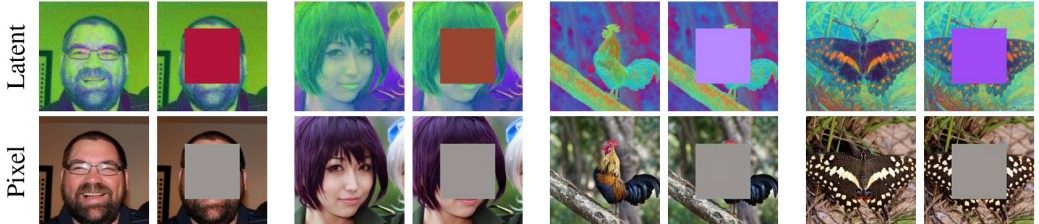

Figure 3: Latent-space masking and its correspondence to pixel space using a central square mask. The encoder and decoder of Stable Diffusion 3.5 (medium) were used. The first row shows latent images alongside the encoded mask applied to each, while the second row shows their decoded counterparts. Notice that the masked regions in the latent space translate directly to analogous masked regions in pixel space. For that sake of visualization, since the latent images have 16 channels, we apply PCA and visualize the first 3 components.

# A   METHOD DETAILS

## A.1   DERIVATION OF THE POSTERIOR (3.1)

Recall that given $\mathbf{z}_s$, the posterior transition of interest is

$$\hat{\pi}_{s|t}^{\theta}(\mathbf{x}_s|\mathbf{z}_s,\mathbf{x}_t,\mathbf{y}) \propto \hat{\ell}_s^{\theta}(\mathbf{y}|\mathbf{x}_s,\mathbf{z}_s)\, p_{s|t}^{\eta,\theta}(\mathbf{x}_s|\mathbf{x}_t)\;.$$

Denoting by $\tilde{\mathbf{y}}_s = \alpha_s\mathbf{y} + \sigma_s\hat{\mathbf{x}}_1^{\theta}(\mathbf{z}_s,s)[\overline{\mathbf{m}}]$ the effective observation, we have that

$$\hat{\ell}_s^{\theta}(\mathbf{y}|\mathbf{x}_s,\mathbf{z}_s) \propto \mathrm{N}(\tilde{\mathbf{y}}_s;\mathbf{x}_s[\overline{\mathbf{m}}],\alpha_s^2\sigma_{\mathbf{y}}^2\mathrm{I}_{d_{\mathbf{y}}})\;,$$

and since the reverse transition writes

$$p_{s|t}^{\eta,\theta}(\mathbf{x}_s|\mathbf{x}_t) = \mathrm{N}(\mathbf{x}_s[\mathbf{m}];\mu_{s|t}^{\theta}(\mathbf{x}_t;\eta)[\mathbf{m}],\eta_s^2\mathrm{I}_{d-d_{\mathbf{y}}})\mathrm{N}(\mathbf{x}_s[\overline{\mathbf{m}}];\mu_{s|t}^{\theta}(\mathbf{x}_t;\eta)[\overline{\mathbf{m}}],\eta_s^2\mathrm{I}_{d_{\mathbf{y}}})\;,$$

we obtain

$$\hat{\pi}_{s|t}^{\theta}(\mathbf{x}_s|\mathbf{z}_s,\mathbf{x}_t,\mathbf{y}) = \mathrm{N}(\mathbf{x}_s[\mathbf{m}];\mu_{s|t}^{\theta}(\mathbf{x}_t;\eta)[\mathbf{m}],\eta_s^2\mathrm{I}_{d-d_{\mathbf{y}}})$$
$$\times \frac{\mathrm{N}(\tilde{\mathbf{y}}_s;\mathbf{x}_s[\overline{\mathbf{m}}],\alpha_s^2\sigma_{\mathbf{y}}^2\mathrm{I}_{d_{\mathbf{y}}})\mathrm{N}(\mathbf{x}_s[\overline{\mathbf{m}}];\mu_{s|t}^{\theta}(\mathbf{x}_t;\eta)[\overline{\mathbf{m}}],\eta_s^2\mathrm{I}_{d_{\mathbf{y}}})}{\int \mathrm{N}(\tilde{\mathbf{y}}_s;\tilde{\mathbf{x}}_s[\overline{\mathbf{m}}],\alpha_s^2\sigma_{\mathbf{y}}^2\mathrm{I}_{d_{\mathbf{y}}})\mathrm{N}(\tilde{\mathbf{x}}_s[\overline{\mathbf{m}}];\mu_{s|t}^{\theta}(\mathbf{x}_t;\eta)[\overline{\mathbf{m}}],\eta_s^2\mathrm{I}_{d_{\mathbf{y}}})\,\mathrm{d}\tilde{\mathbf{x}}_s[\overline{\mathbf{m}}]}\;.$$

The formula (3.4) follows by applying Bishop (2006, equation 2.116) to the second normalized transition on the right-hand side.

## A.2   COMPARISON WITH RELATED WORKS

We start by providing an explicit comparison with the closest works.

**Comparison with the transition in Cardoso et al. (2023).**   Let $\tau \in [0,1]$ be a timestep such that $\sigma_{\mathbf{y}} = \sigma_\tau/\alpha_\tau$. Such a $\tau$ always exists when the linear schedule is used for example. The transition used in the SMC algorithm in Cardoso et al. (2023) for $s > \tau$ is given by

$$\hat{\pi}_{s|t}^{\theta}(\mathbf{x}_s|\mathbf{x}_t) \propto \mathrm{N}(\alpha_s\mathbf{y};\mathbf{x}_s[\overline{\mathbf{m}}],\sigma_{s|\tau}^2\mathrm{I}_{d_{\mathbf{y}}})p_{s|t}^{\eta,\theta}(\mathbf{x}_s|\mathbf{x}_t) \tag{A.1}$$

Using the same conjugation formulas as in the previous section, we find that

$$\hat{\pi}_{s|t}^{\theta}(\mathbf{x}_s|\mathbf{x}_t) = \mathrm{N}(\mathbf{x}_s[\mathbf{m}],\mu_{s|t}^{\theta}(\mathbf{x}_t;\eta)[\mathbf{m}],\eta_s^2\mathrm{I}_{d-d_{\mathbf{y}}})$$
$$\times \mathrm{N}(\mathbf{x}_s[\overline{\mathbf{m}}],(1-\tilde{\gamma}_{s|t})\mu_{s|t}^{\theta}(\mathbf{x}_t;\eta)[\overline{\mathbf{m}}]+\tilde{\gamma}_{s|t}\alpha_s\mathbf{y},\sigma_{t|\tau}^2\tilde{\gamma}_{s|t}\mathrm{I}_{d-d_{\mathbf{y}}}),\quad \text{(A.2)}$$

where $\sigma_{t|\tau}^2 := \sigma_t^2 - (\alpha_t/\alpha_\tau)^2\sigma_\tau^2$ and $\tilde{\gamma}_{s|t} = \eta_s^2/(\eta_s^2+\sigma_{t|\tau}^2)$. This is to be contrasted with our update, given a sample $\mathbf{z}_s$,

$$\hat{\pi}_{s|t}^{\theta}(\mathbf{x}_s|\mathbf{z}_s,\mathbf{x}_t,\mathbf{y}) = \mathrm{N}\big(\mathbf{x}_s[\mathbf{m}];\boldsymbol{\mu}_{s|t}^{\theta}(\mathbf{x}_t;\eta)[\mathbf{m}],\eta_s^2\mathrm{I}_{d-d_{\mathbf{y}}}\big)$$
$$\times \mathrm{N}\Big(\mathbf{x}_s[\overline{\mathbf{m}}];(1-\gamma_{s|t})\boldsymbol{\mu}_{s|t}^{\theta}(\mathbf{x}_t;\eta)[\overline{\mathbf{m}}]+\gamma_{s|t}\big(\alpha_s\mathbf{y}+\sigma_s\hat{\mathbf{x}}_1^{\theta}(\mathbf{z}_s,s)[\overline{\mathbf{m}}]\big),\alpha_s^2\sigma_{\mathbf{y}}^2\gamma_{s|t}\,\mathrm{I}_{d_{\mathbf{y}}}\Big)\;,$$

where $\gamma_{s|t} := \eta_s^2/(\eta_s^2+\alpha_s^2\sigma_{\mathbf{y}}^2)$. Hence, MCGDIFF differs from DING on the choice of effective observation, which in this case is $\tilde{\mathbf{y}}_s = \alpha_s\mathbf{y}$, the choice of variance in the transition and the coefficient of the convex combination. From (A.1) it can be seen that MCGDIFF assumes the approximate model $\mathrm{N}(\alpha_s\mathbf{y};\mathbf{x}_s[\overline{\mathbf{m}}],\sigma_{s|\tau}^2\mathrm{I}_{d_{\mathbf{y}}})$ for the true likelihood $\ell_s(\mathbf{y}|\mathbf{x}_s)$ (2.7).

**Comparison with the transition in Zhu et al. (2023) and Martin et al. (2025).** We now write explicitly the algorithm PNP-FLOW (Martin et al., 2025, Algorithm 3) adapted to the inpainting problem we consider; see Algorithm 2. We have simply adapted the notations and used $F(\mathbf{x}) = \|\mathbf{y} - \mathbf{x}[\overline{\mathbf{m}}]\|^2/(2\sigma_{\mathbf{y}}^2)$ in Martin et al. (2025, Algorithm 3). Thus, the transition used in Algorithm 2 is

$$\hat{\pi}_{s|t}^{\theta}(\mathbf{x}_s|\mathbf{x}_t) \propto \mathrm{N}(\mathbf{x}_s[\mathbf{m}], \alpha_s \hat{\mathbf{x}}_0^{\theta}(\mathbf{x}_t, t)[\mathbf{m}], \sigma_s^2 \mathrm{I}_{d-d_{\mathbf{y}}})$$
$$\times \mathrm{N}\left(\mathbf{x}_s[\overline{\mathbf{m}}], \left(1 - \frac{\gamma_s}{\sigma_{\mathbf{y}}^2}\right)\alpha_s \hat{\mathbf{x}}_0^{\theta}(\mathbf{x}_t, t)[\overline{\mathbf{m}}] + \frac{\gamma_s}{\sigma_{\mathbf{y}}^2}\alpha_s\mathbf{y}, \sigma_s^2 \mathrm{I}_{d-d_{\mathbf{y}}}\right) .$$

In the case of the DDIM schedule $\eta_s = \sigma_s$, we have that $\mu_{s|t}^{\theta}(\mathbf{x}_t) = \alpha_s \hat{\mathbf{x}}_0^{\theta}(\mathbf{x}_t, t)$, and the MCGDIFF transition (Cardoso et al., 2023) in (A.2) writes

$$\hat{\pi}_{s|t}^{\theta}(\mathbf{x}_s|\mathbf{x}_t) = \mathrm{N}(\mathbf{x}_s[\mathbf{m}], \alpha_s \hat{\mathbf{x}}_0^{\theta}(\mathbf{x}_t, t)[\mathbf{m}], \sigma_s^2 \mathrm{I}_{d-d_{\mathbf{y}}})$$
$$\times \mathrm{N}(\mathbf{x}_s[\overline{\mathbf{m}}], (1 - \tilde{\gamma}_{s|t})\alpha_s \hat{\mathbf{x}}_0^{\theta}(\mathbf{x}_t, t)[\overline{\mathbf{m}}] + \tilde{\gamma}_{s|t}\alpha_s\mathbf{y}, \sigma_{s|\tau}^2 \tilde{\gamma}_{s|t}\mathrm{I}_{d-d_{\mathbf{y}}}).$$

Hence, the main difference lies in the coefficient of the convex combination and the variance used.

---

**Algorithm 2** PNP-FLOW reinterpreted

---

1: **Input:** Decreasing timesteps $(t_k)_{k=K}^0$ with $t_K = 1$, $t_0 = 0$; adaptive stepsizes $(\gamma_k)_{k=K}^0$.
2: **Initialize:** $\hat{\mathbf{x}}_0 \in \mathbb{R}^d$.
3: **for** $k = K - 1$ **to** $1$ **do**
4: $\quad$ $\hat{\mathbf{x}}_0[\overline{\mathbf{m}}] \leftarrow (1 - \frac{\gamma_k}{\sigma_{\mathbf{y}}^2})\hat{\mathbf{x}}_0[\overline{\mathbf{m}}] + \frac{\gamma_k}{\sigma_{\mathbf{y}}^2}\mathbf{y}$
5: $\quad$ $\mathbf{w} \sim \mathcal{N}(0_d, \mathrm{I}_d)$
6: $\quad$ $\mathbf{x} \leftarrow \alpha_{t_k}\hat{\mathbf{x}}_0 + \sigma_{t_k}\mathbf{w}$
7: $\quad$ $\hat{\mathbf{x}}_0 \leftarrow \hat{\mathbf{x}}_0^{\theta}(\mathbf{x}, t_k)$
8: **end for**
9: **Return:** $\hat{\mathbf{x}}_0$

---

**Comparison with the transition in Kim et al. (2025); Patel et al. (2024).** Here we explicitly write the transition of FLOWDPS for the inpainting case in order to understand the main differences without our method. For this purpose we rewrite Kim et al. (2025, Algorithm 1) using our notations. We note that the algorithm is written for the linear schedule $\alpha_t = 1 - t$, $\sigma_t = t$ and the choice of DDIM schedule $\eta_t = \sigma_t\sqrt{1 - \sigma_t}$, but we still write it with general notations to streamline the comparison with Algorithm 1. We also assume for the sake of simplicity that the optimization problem is solved exactly in Kim et al. (2025, line 7) (since there is no decoder as we solve the inverse problem in the latent space). The algorithm is given in Algorithm 3. In the specific setting where the linear schedule is used, setting $\gamma_k = \sigma_{\mathbf{y}}^2 \sigma_{t_k}$ in Algorithm 2 recovers Algorithm 3 when $\eta_k = \sigma_{t_k}$. Finally, we note that FLOWDPS can be understood as a noisy version of the FLOWCHEF algorithm (Patel et al., 2024) and overall, follows the line of work of methods that learn a residual that is then used to translate the denoiser (Bansal et al., 2023; Zhu et al., 2023).

**Comparison with DiffPIR (Zhu et al., 2023) and DDNM (Wang et al., 2023b).** We provide the DIFFPIR algorithm (Zhu et al., 2023, Algorithm 1) adapted to our inpainting case using our own notation in Algorithm 4. In Line 6 we write the exact solution to the optimization problem in the original algorithm. We write the associated transition in a convenient form that allows a seamless comparison with our algorithm. Define $\gamma_t := \sigma_t^2/(\sigma_t^2 + \lambda\alpha_t^2\sigma_{\mathbf{y}}^2)$. Then, the transition

$$\hat{\pi}_{s|t}^{\theta}(\mathbf{x}_s|\mathbf{x}_t, \mathbf{y}) = \mathrm{N}\big(\mathbf{x}_s[\mathbf{m}]; \boldsymbol{\mu}_{s|t}^{\theta}(\mathbf{x}_t;\eta)[\mathbf{m}], \eta_s^2 \mathrm{I}_{d-d_{\mathbf{y}}}\big)$$
$$\times \mathrm{N}\Big(\mathbf{x}_s[\overline{\mathbf{m}}]; (1 - \gamma_t)\boldsymbol{\mu}_{s|t}^{\theta}(\mathbf{x}_t;\eta)[\overline{\mathbf{m}}] + \gamma_t\big(\alpha_s\mathbf{y} + (\sigma_s^2 - \eta_s^2)^{1/2}\frac{\mathbf{x}_t[\overline{\mathbf{m}}] - \alpha_t\mathbf{y}}{\sigma_t}\big), \eta_s^2 \mathrm{I}_{d_{\mathbf{y}}}\Big) ,$$

corresponds to one step of Algorithm 4. We highlight key distinctions:

- Setting $\eta_s^2 = \sigma_s^2$ recovers the same transition as in PNP-FLOW.
- The main distinction lies in the mean of the Gaussian transition for the unmasked region: it is a convex combination of $\boldsymbol{\mu}^{\theta}(\mathbf{x}_t;\eta)[\overline{\mathbf{m}}]$ and an effective observation $\alpha_s\mathbf{y} + (\sigma_s^2 - \eta_s^2)^{1/2}(\mathbf{x}_t[\overline{\mathbf{m}}] -$

---

**Algorithm 3** FLOWDPS reinterpreted

---

1: **Input:** decreasing timesteps $(t_k)_{k=K}^0$ with $t_K = 1$, $t_0 = 0$; original image $\mathbf{x}_*$; mask $\mathbf{m}$; DDIM parameters $(\eta_k)_{k=K}^0$.
2: $\mathbf{y} \leftarrow \mathbf{x}_*[\overline{\mathbf{m}}]$
3: $\mathbf{x} \sim \mathcal{N}(0, \mathrm{I}_d)$
4: **for** $k = K - 1$ **to** $0$ **do**
5: $\quad \hat{\mathbf{x}}_0 \leftarrow \mathbf{x}_0^\theta(\mathbf{x}, t_{k+1})$
6: $\quad \hat{\mathbf{x}}_1 \leftarrow (\mathbf{x} - \alpha_{t_{k+1}} \hat{\mathbf{x}}_0) / \sigma_{t_{k+1}}$
7: $\quad \textcolor{red}{\hat{\mathbf{x}}_0[\overline{\mathbf{m}}] \leftarrow \alpha_{t_k} \hat{\mathbf{x}}_0[\overline{\mathbf{m}}] + \sigma_{t_k} \mathbf{y}}$
8: $\quad \boldsymbol{\mu} \leftarrow \alpha_{t_k} \hat{\mathbf{x}}_0 + (\sigma_{t_k}^2 - \eta_k^2)^{1/2} \hat{\mathbf{x}}_1$
9: $\quad \mathbf{w} \sim \mathcal{N}(0_d, \mathrm{I}_d)$
10: $\quad \mathbf{x} \leftarrow \boldsymbol{\mu} + \eta_k \mathbf{w}$
11: **end for**
12: **Return:** $\mathbf{x}$

---

$\alpha_t \mathbf{y}) / \sigma_t$. In our algorithm, the effective observation instead takes the form $\alpha_s \mathbf{y} + \sigma_s \hat{\mathbf{x}}_1^\theta(\mathbf{x}_s, s)[\overline{\mathbf{m}}]$. We estimate the residual noise using the pre-trained model at timestep $s$, whereas DIFFPIR computes it as $(\mathbf{x}_t[\overline{\mathbf{m}}] - \alpha_t \mathbf{y}) / \sigma_t$.

- This residual noise is scaled differently: by $(\sigma_s^2 - \eta_s^2)^{1/2}$ in DIFFPIR, and by $\sigma_s$ in our method.
- The convex combination coefficient in our cases is $\gamma_{s|t} = \eta_s^2 / (\eta_s^2 + \alpha_s^2 \sigma_{\mathbf{y}}^2)$ whereas for DIFFPIR it is set to $\gamma_t = \sigma_t^2 / (\sigma_t^2 + \alpha_t^2 \sigma_{\mathbf{y}}^2)$.
- Finally, the noise-free ($\sigma_{\mathbf{y}} = 0$) version of DIFFPIR recovers the DDNM algorithm (Zhang et al., 2023).

---

**Algorithm 4** DIFFPIR reinterpreted

---

1: **Input:** Decreasing timesteps $(t_k)_{k=K}^0$ with $t_K = 1$, $t_0 = 0$; scaling $\lambda$; original image $\mathbf{x}_*$; mask $\mathbf{m}$; DDIM parameters $(\eta_k)_{k=K}^0$
2: $\mathbf{y} \leftarrow \mathbf{x}_*[\overline{\mathbf{m}}]$
3: $\mathbf{x} \sim \mathcal{N}(0, \mathrm{I}_d)$.
4: **for** $k = K - 1$ **to** $1$ **do**
5: $\quad \hat{\mathbf{x}}_0 \leftarrow \mathbf{x}_0^\theta(\mathbf{x}, t_{k+1})$
6: $\quad \textcolor{red}{\hat{\mathbf{x}}_0[\overline{\mathbf{m}}] \leftarrow \frac{\sigma_{t_{k+1}}^2}{\sigma_{t_{k+1}}^2 + \lambda \sigma_{\mathbf{y}}^2 \alpha_{t_{k+1}}^2} \mathbf{y} + \frac{\lambda \sigma_{\mathbf{y}}^2 \alpha_{t_{k+1}}^2}{\sigma_{t_{k+1}}^2 + \lambda \sigma_{\mathbf{y}}^2 \alpha_{t_{k+1}}^2} \hat{\mathbf{x}}_0[\overline{\mathbf{m}}]}$
7: $\quad \hat{\mathbf{x}}_1 \leftarrow (\mathbf{x} - \alpha_{t_{k+1}} \hat{\mathbf{x}}_0) / \sigma_{t_{k+1}}$
8: $\quad \mathbf{w} \sim \mathcal{N}(0_d, \mathrm{I}_d)$
9: $\quad \mathbf{x} \leftarrow \alpha_{t_k} \hat{\mathbf{x}}_0 + (\sigma_{t_k}^2 - \eta_k^2)^{1/2} \hat{\mathbf{x}}_1 + \eta_k \mathbf{w}$
10: **end for**
11: **Return:** $\mathbf{x}$

---

**Further related methods.** Here we continue our discussion of VJP-free methods. The DAPS algorithm (Zhang et al., 2025) proposes sampling, given the previous state $X_{t_{k+1}}$, a clean state $\hat{X}_0$ by performing Langevin Monte Carlo steps on the posterior distribtion $\pi_{0|t_{k+1}}(\cdot | X_{t_{k+1}}, \mathbf{y})$. This step is performed approximately by replacing the prior transition $p_{0|t_{k+1}}(\cdot | X_{t_{k+1}})$ with a Gaussian approximation centered at the denoiser $\hat{\mathbf{x}}_0^\theta(X_{t_{k+1}}, t_{k+1})$. Then, given $\hat{X}_0$, the next state $X_{t_k}$ is drawn from $\mathcal{N}(\alpha_{t_k} \hat{X}_0, \sigma_{t_k}^2 \mathrm{I}_d)$.

One important aspect of our method is that we circumvent differentiation through the denoiser but also the decoder, as the diffusion models we consider operate in the latent space. We do so by downsampling the mask into the latent space. In contrast, the recent work of Spagnoletti et al. (2025) also circumvents differentiation through the denoiser, but does so by lifting the latent states into pixel space and optimizing the likelihood there. The result of the optimization is then projected back into the latent space and then undergoes back-and-forth noise-denoising steps.

Finally, several recent works (Mardani et al., 2024; Zilberstein et al., 2025; Erbach et al., 2025) adopt a variational perspective: the target distribution is approximated by a Gaussian distribution whose

parameters are iteratively estimated by minimizing a combination of an observation-fidelity loss and a score-matching-like loss.

*VJP-based methods.* A broad class of zero-shot approaches builds on the guidance approximation (2.8) to estimate $\nabla_{\mathbf{x}_t} \log \ell_t(\mathbf{y}|\mathbf{x}_t)$. Song et al. (2023a) approximate $p_{0|t}$ by a Gaussian with mean $\hat{\mathbf{x}}_0^\theta(\cdot, t)$ and a tuned covariance. For the inpainting setting in (2.4), plugging this approximation into (2.7) yields an integral that can be computed in a closed form, providing a proxy for $\ell_t(\mathbf{y}|\cdot)$. Several works exploit the link between the covariance of $p_{0|t}(\cdot|\mathbf{x}_t)$ and the Jacobian of the denoiser (Meng et al., 2021). This observation underpins the methods of Finzi et al. (2023), Stevens et al. (2023), and Boys et al. (2023), which derive likelihood scores by estimating or inverting the Jacobian. These approaches require solving large linear systems and backpropagating through the denoiser, both computationally expensive operations. To reduce cost, these works assume a locally constant Jacobian around $\mathbf{x}_t$, but updates still involve either explicit matrix inversion or repeated VJPs. In practice, diagonal approximations based on row sums are commonly used to approximate the covariance matrix (Boys et al., 2023), or conjugate gradient methods are employed to circumvent the need for full matrix inversion (Rozet et al., 2024). For general likelihoods $\ell_0(\mathbf{y}|\cdot)$, Song et al. (2023b) combine the Gaussian posterior model of Song et al. (2023a) with Monte Carlo sampling to approximate $\ell_t(\mathbf{y}|\cdot)$. In the latent setting, Rout et al. (2024b) apply the DPS approximation jointly with a regularizer that encourages latent variables to remain near encoder–decoder fixed points. Other methods modify the sampling dynamics. Moufad et al. (2025) propose a two-stage procedure: the chain is first moved to an earlier time $\ell \ll t_k$, where the DPS approximation is applied to sample from an approximate conditional at $\ell$, before returning to $t_k$ via additional noising steps. Janati et al. (2025a) incorporate a related idea into a Gibbs sampling framework. Overall, these methods remain fundamentally VJP-based and inherit substantial memory and runtime overhead from repeated backpropagation through the denoiser. By contrast, our decoupled guidance relies exclusively on forward denoiser evaluations and closed-form Gaussian updates, thereby eliminating VJPs entirely while retaining competitive performance.

For a complete review of zero-shot posterior sampling methods see Daras et al. (2024); Janati et al. (2025b); Chung et al. (2025).

### A.3  BEHAVIOR UNDER INCREASED RUNTIME.

We extend the ablation study in Section 4.2 by examining the behavior of DING when the number of NFEs is increased. Specifically, we vary the budget from 20 to 500 NFEs on the DIV2K dataset and report results across different masking patterns; see Figure 4. All metrics improve steadily as the budget grows, reaching their best values around 200 NFEs (10s runtime). Beyond this point, performance saturates and exhibits a slight degradation at 500 NFEs. These results suggest that our default DDIM schedule is well suited to low and mid-NFE regimes—which are most relevant for practical settings—but may not be fully optimized for larger budgets.

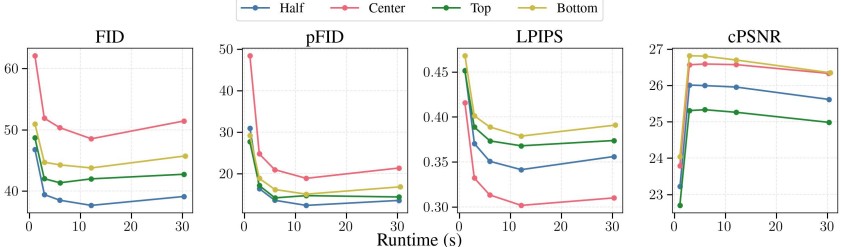

Figure 4: Performance of DING on DIV2K under varying NFE budgets (20 to 500) across different masking patterns. Runtimes are measured on a H100 GPU.

### A.4  LIMITATION

We observed that the quality of reconstructions is highly sensitive to the specificity of the textual prompt. When the prompt is under-specified or lacks sufficient semantic detail, the resulting samples may exhibit reduced coherence, particularly in large masked regions where contextual consistency

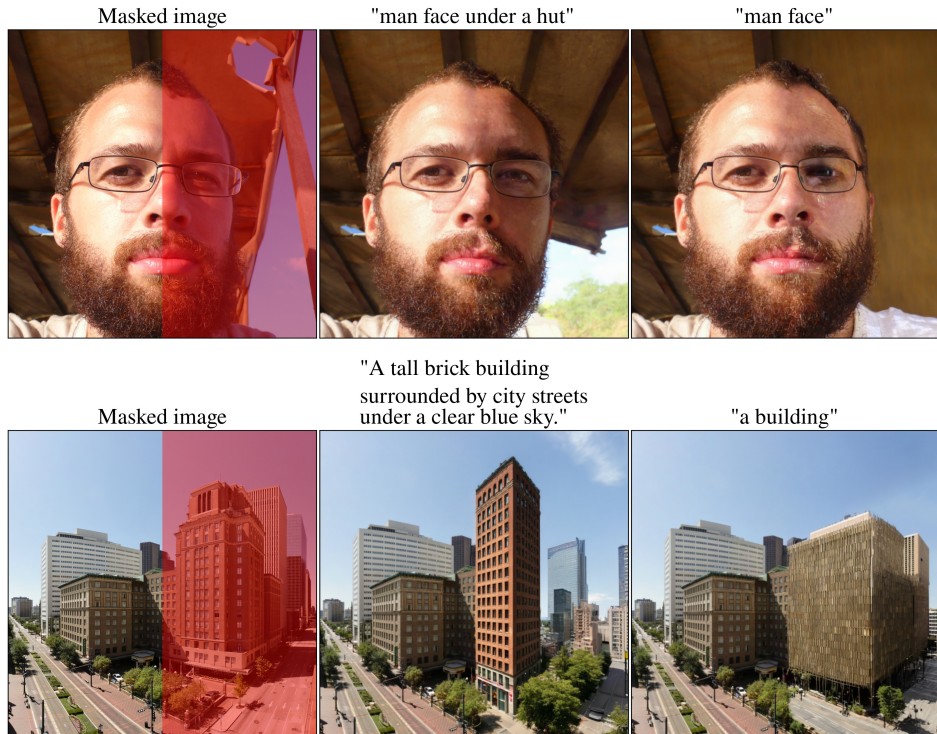

Figure 5: Effect of prompt precision on inpainting quality.

is critical. This issue manifests as mismatched textures or backgrounds, or inconsistent object boundaries, even when the visible area is faithfully preserved. To illustrate this behavior, we compare reconstructions obtained with well-defined prompts against those generated using vague or ambiguous ones. Examples are provided in Figure 5 and 6.

### A.5    BIAS IN GAUSSIAN CASE

For the sake of simplicity we assume that $p_0 \coloneqq \mathcal{N}(0_d, \Sigma)$ where $\Sigma$ is a covariance matrix. We also write the likelihood as $\ell_0(\mathbf{y}|\mathbf{x}_0) = \mathrm{N}(\mathbf{y}; P_{\overline{\mathbf{m}}}\mathbf{x}_0, \sigma_{\mathbf{y}}^2 \mathrm{I}_{d_{\mathbf{y}}})$ where $P_{\overline{\mathbf{m}}} \in \mathbb{R}^{d_{\mathbf{y}} \times d}$ is the matrix satisfying $P_{\overline{\mathbf{m}}}\mathbf{x} = \mathbf{x}[\overline{\mathbf{m}}]$. Define $D_t \coloneqq \alpha_t \Sigma (\alpha_t^2 \Sigma + \sigma_t^2 \mathrm{I}_d)^{-1}$. Then, the denoiser and noise predictors are given by

$$\hat{\mathbf{x}}_0(\mathbf{x}_t, t) = D_t \mathbf{x}_t \ , \quad \hat{\mathbf{x}}_1(\mathbf{x}_t, t) = \sigma_t^{-1} \left( \mathrm{I}_d - \alpha_t D_t \right) \mathbf{x}_t \ .$$

We consider hereafter the DDIM transitions $p_{s|t}^\eta(\mathbf{x}_t|\mathbf{x}_t) \coloneqq \mathrm{N}(\mathbf{x}_s; \boldsymbol{\mu}_{s|t}(\mathbf{x}_t; \eta), \eta_s^2 \mathrm{I}_d)$ where

$$\boldsymbol{\mu}_{s|t}(\mathbf{x}_t; \eta) \coloneqq \alpha_s \hat{\mathbf{x}}_0(\mathbf{x}_t, t) + \sqrt{\sigma_s^2 - \eta_s^2} \hat{\mathbf{x}}_1(\mathbf{x}_t, t)$$

In this section we analyze the bias of the DING one-step transition relative to the posterior transition involving the DPS likelihood (2.8); *i.e.* we compare the transition

$$\hat{\pi}_{s|t}^{\mathsf{ding}}(\mathbf{x}_s|\mathbf{x}_t, \mathbf{y}) \ \coloneqq \ \mathbb{E}\left[ \hat{\pi}_{s|t}^{\mathsf{ding}}(\mathbf{x}_s|Z_s, \mathbf{x}_t, \mathbf{y}) \right] \ , \tag{A.3}$$

where $Z_s \sim p_{s|t}^\eta(\cdot|\mathbf{x}_t)$ and

$$\hat{\pi}_{s|t}^{\mathsf{ding}}(\mathbf{x}_s|\mathbf{z}_s, \mathbf{x}_t, \mathbf{y}) \propto \ell_0(\mathbf{y}|\frac{\mathbf{x}_s - \sigma_s \hat{\mathbf{x}}_1(\mathbf{z}_s, s)}{\alpha_s}) p_{s|t}^\eta(\mathbf{x}_s|\mathbf{x}_t)$$

against

$$\hat{\pi}_{s|t}^{\mathsf{dps}}(\mathbf{x}_s|\mathbf{x}_t, \mathbf{y}) \propto \ell_0(\mathbf{y}|\hat{\mathbf{x}}_0(\mathbf{x}_s, s)) p_{s|t}^\eta(\mathbf{x}_s|\mathbf{x}_t) \ .$$

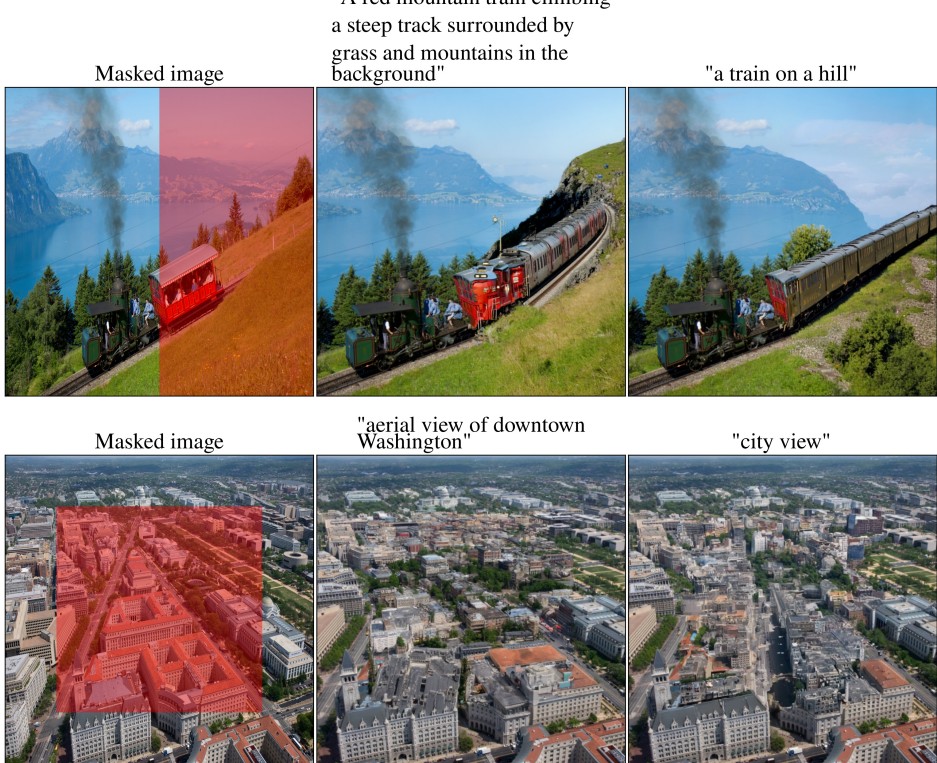

Figure 6: Effect of prompt precision on inpainting quality.

We define $M := P_{\overline{\mathbf{m}}}^\top P_{\overline{\mathbf{m}}}$, which is an orthogonal projection matrix since $M^\top = M$, $P_{\overline{\mathbf{m}}} P_{\overline{\mathbf{m}}}^\top = \mathrm{I}_{d_{\mathbf{y}}}$, and thus $M^2 = M$. We also introduce the quantity

$$\varepsilon_s := \|(D_s^\top - \alpha_s^{-1}\mathrm{I}_d)M\|_{\mathrm{op}},$$

which quantifies how far the Jacobian of the denoiser $\hat{\mathbf{x}}_0(\cdot, s)$ deviates from the Jacobian of the DING denoiser approximation *on the observed coordinates*. In the following proposition, we characterize the asymptotic behavior of the DPS and DING posterior transition means and covariances as $\eta_s \to 0$, and we express the mean bias in terms of $\varepsilon_s$. In Proposition 2, we also provide an explicit upper bound on $\varepsilon_s$ in terms of the schedule and the minimum eigenvalue of the prior covariance $\Sigma$.

**Proposition 1.** *Both* $\hat{\pi}_{s|t}^{\mathsf{dps}}(\cdot|\mathbf{x}_t, \mathbf{y})$ *and* $\hat{\pi}_{s|t}^{\mathsf{ding}}(\cdot|\mathbf{x}_t, \mathbf{y})$ *are Gaussian distributions with mean and covariance respectively* $(\boldsymbol{\mu}_{s|t}^{\mathsf{dps}}(\mathbf{x}_t, \mathbf{y}), \Sigma_{s|t}^{\mathsf{dps}})$ *and* $(\boldsymbol{\mu}_{s|t}^{\mathsf{ding}}(\mathbf{x}_t, \mathbf{y}), \Sigma_{s|t}^{\mathsf{ding}})$ *satisfying*

$$\|\Sigma_{s|t}^{\mathsf{dps}} - \Sigma_{s|t}^{\mathsf{ding}}\| = \mathcal{O}(\eta_s^4)$$

*and*

$$\|\boldsymbol{\mu}_{s|t}^{\mathsf{dps}}(\mathbf{x}_t, \mathbf{y}) - \boldsymbol{\mu}_{s|t}^{\mathsf{ding}}(\mathbf{x}_t, \mathbf{y})\| = \mathcal{O}\left(\eta_s^2\big(\varepsilon_s(\|\mathbf{y}\| + \|M\boldsymbol{\mu}_{s|t}(\mathbf{x}_t; \eta)\|) + \varepsilon_s^2\|\boldsymbol{\mu}_{s|t}(\mathbf{x}_t; \eta)\|\big)\right).$$

*as* $\eta_s \to 0$.

*Proof.* Using the standard Gaussian conjugation formula (Bishop, 2006, equation 2.116), we have that $\hat{\pi}_{s|t}^{\mathsf{dps}}(\mathbf{x}_s|\mathbf{x}_t, \mathbf{y}) = \mathrm{N}(\mathbf{x}; \mathbf{m}_{s|t}^{\mathsf{dps}}(\mathbf{x}_t, \mathbf{y}), \Sigma_s^{\mathsf{dps}})$ with

$$\mathbf{m}_{s|t}^{\mathsf{dps}}(\mathbf{x}_t, \mathbf{y}) := \Sigma_{s|t}^{\mathsf{dps}}(\eta_s^{-2}\boldsymbol{\mu}_{s|t}(\mathbf{x}_t; \eta) + \sigma_{\mathbf{y}}^{-2}D_s^\top P_{\overline{\mathbf{m}}}^\top \mathbf{y}),$$

$$\Sigma_{s|t}^{\mathsf{dps}} := \big(\eta_s^{-2}\mathrm{I}_d + \sigma_{\mathbf{y}}^{-2}(P_{\overline{\mathbf{m}}}D_s)^\top P_{\overline{\mathbf{m}}}D_s\big)^{-1}.$$

Next, for the DING transition, first set $b_s(Z_s) := -(\sigma_s/\alpha_s)P_{\overline{\mathbf{m}}}\hat{\mathbf{x}}_1(Z_s, s)$. Gaussian conjugacy with $p_{s|t}^\eta(\mathbf{x}_s|\mathbf{x}_t) = \mathcal{N}(\mathbf{x}_s; \boldsymbol{\mu}_{s|t}(\mathbf{x}_t;\eta), \eta_s^2 I_d)$ gives

$$\hat{\pi}_{s|t}^{\text{ding}}(\mathbf{x}_s|Z_s, \mathbf{x}_t, \mathbf{y}) = N\Big(\mathbf{x}_s; \widetilde{\Sigma}_{s|t}^{\text{ding}}\big(\eta_s^{-2}\boldsymbol{\mu}_{s|t}(\mathbf{x}_t;\eta) + \sigma_{\mathbf{y}}^{-2}\alpha_s^{-1}P_{\overline{\mathbf{m}}}^\top(\mathbf{y} - b_s(Z_s))\big), \widetilde{\Sigma}_{s|t}^{\text{ding}}\Big),$$

and $\widetilde{\Sigma}_{s|t}^{\text{ding}} := \big(\eta_s^{-2}I_d + \alpha_s^{-2}\sigma_{\mathbf{y}}^{-2}P_{\overline{\mathbf{m}}}^\top P_{\overline{\mathbf{m}}}\big)^{-1}$ . Since the mean of this conditional distribution is clearly affine in $Z_s$, we integrate it out, yielding that $\hat{\pi}_{s|t}^{\text{ding}}(\mathbf{x}_s|\mathbf{x}_t, \mathbf{y}) = N(\mathbf{x}_s; \mathbf{m}_{s|t}^{\text{ding}}(\mathbf{x}_t, \mathbf{y}), \Sigma_{s|t}^{\text{ding}})$ where

$$\mathbf{m}_{s|t}^{\text{ding}}(\mathbf{x}_t, \mathbf{y}) := \widetilde{\Sigma}_{s|t}^{\text{ding}}\big(\eta_s^{-2}\boldsymbol{\mu}_{s|t}(\mathbf{x}_t;\eta) + \sigma_{\mathbf{y}}^{-2}\alpha_s^{-1}P_{\overline{\mathbf{m}}}^\top\mathbf{y}$$
$$+ (\sigma_{\mathbf{y}}^{-2}\alpha_s^{-2})P_{\overline{\mathbf{m}}}^\top P_{\overline{\mathbf{m}}}(I_d - \alpha_s D_s)\boldsymbol{\mu}_{s|t}(\mathbf{x}_t;\eta)\big) ,$$

$$\Sigma_{s|t}^{\text{ding}} := \widetilde{\Sigma}_{s|t}^{\text{ding}} + \frac{\eta_s^2}{\sigma_{\mathbf{y}}^4\alpha_s^4}\widetilde{\Sigma}_{s|t}^{\text{ding}}P_{\overline{\mathbf{m}}}^\top P_{\overline{\mathbf{m}}}(I_d - \alpha_s D_s)(\widetilde{\Sigma}_{s|t}^{\text{ding}}P_{\overline{\mathbf{m}}}^\top P_{\overline{\mathbf{m}}}(I_d - \alpha_s D_s))^\top .$$

**Small-noise regime.** We now study the behavior of both transitions when the DDIM kernel variance $\eta_s^2$ tends to zero. For simplicity we define

$$K_{\text{dps}} := \sigma_{\mathbf{y}}^{-2}D_s^\top P_{\overline{\mathbf{m}}}^\top P_{\overline{\mathbf{m}}}D_s , \quad K_{\text{ding}} := \alpha_s^{-2}\sigma_{\mathbf{y}}^2 M$$

and $R_s = I_d - \alpha_s D_s$, $M = P_{\overline{\mathbf{m}}}^\top P_{\overline{\mathbf{m}}}$. Then,

$$\Sigma_{s|t}^{\text{dps}} = (\eta_s^{-2}I_d + K_{\text{dps}})^{-1} ,$$

$$\Sigma_{s|t}^{\text{ding}} = (\eta_s^{-2}I_d + K_{\text{ding}})^{-1} + \frac{\eta_s^2}{\alpha_s^4\sigma_y^4}(\eta_s^{-2}I_d + K_{\text{ding}})^{-1}MR_sR_s^\top M(\eta_s^{-2}I_d + K_{\text{ding}})^{-1}.$$

We use throughout that for any fixed matrix $K$, we have that when $\eta_s^2\|K\|_{\text{op}} < 1$,

$$(\eta_s^{-2}I_d + K)^{-1} = \eta_s^2(I_d - \eta_s^2 K) + R_2(\eta_s), \quad \|R_2(\eta_s)\| \le \eta_s^6\frac{\|K\|_{\text{op}}^2}{1 - \eta_s^2\|K\|_{\text{op}}}. \quad (A.4)$$

This follows from the standard Neumann (geometric) series expansion. Applying (A.4) with $\eta_s^2 \le \min(1/\|K_{\text{dps}}\|_{\text{op}}, 1/\|K_{\text{ding}}\|_{\text{op}})$, we get

$$\Sigma_{s|t}^{\text{dps}} = \eta_s^2(I_d - \eta_s^2 K_{\text{dps}}) + \mathcal{O}(\eta_s^6),$$

$$\Sigma_{s|t}^{\text{ding}} = \eta_s^2(I_d - \eta_s^2 K_{\text{ding}}) + \frac{\eta_s^6}{\alpha_s^4\sigma_y^4}MR_sR_s^\top M + \mathcal{O}(\eta_s^6).$$

and thus

$$\Sigma_{s|t}^{\text{dps}} - \Sigma_{s|t}^{\text{ding}} = \mathcal{O}(\eta_s^4) .$$

Plugging these expansions in the mean terms, we find that

$$\mathbf{m}_{s|t}^{\text{dps}}(\mathbf{x}_t, \mathbf{y}) = \boldsymbol{\mu}_{s|t}(\mathbf{x}_t;\eta) + \eta_s^2(\sigma_y^{-2}D_s^\top P_{\overline{\mathbf{m}}}^\top\mathbf{y} - K_{\text{dps}}\boldsymbol{\mu}_{s|t}(\mathbf{x}_t;\eta)) + \mathcal{O}(\eta_s^4),$$

$$\mathbf{m}_{s|t}^{\text{ding}}(\mathbf{x}_t, \mathbf{y}) = \boldsymbol{\mu}_{s|t}(\mathbf{x}_t;\eta) + \eta_s^2(\alpha_s^{-1}\sigma_y^{-2}P_{\overline{\mathbf{m}}}^\top\mathbf{y} + \alpha_s^{-2}\sigma_y^{-2}MR_s\boldsymbol{\mu}_{s|t} - K_{\text{ding}}\boldsymbol{\mu}_{s|t}(\mathbf{x}_t;\eta)) + \mathcal{O}(\eta_s^4).$$

This yields

$$\mathbf{m}_{s|t}^{\text{dps}}(\mathbf{x}_t, \mathbf{y}) - \mathbf{m}_{s|t}^{\text{ding}}(\mathbf{x}_t, \mathbf{y}) = \eta_s^2\sigma_y^{-2}\big[(\alpha_s^{-1}I_d - D_s^\top)P_{\overline{\mathbf{m}}}^\top\mathbf{y}$$
$$+ \alpha_s^{-2}MR_s\boldsymbol{\mu}_{s|t}(\mathbf{x}_t;\eta) - (\alpha_s^{-2}M - D_s^\top MD_s)\boldsymbol{\mu}_{s|t}(\mathbf{x}_t;\eta)\big] + \mathcal{O}(\eta_s^4),$$

We now proceed to further upper bound the leading term. Define $E_s := D_s - \alpha_s^{-1}I_d$. Then $R_s = -\alpha_s E_s$ and we have that

$$\mathbf{m}_{s|t}^{\text{dps}}(\mathbf{x}_t, \mathbf{y}) - \mathbf{m}_{s|t}^{\text{ding}}(\mathbf{x}_t, \mathbf{y}) =$$
$$\eta_s^2\sigma_{\mathbf{y}}^{-2}\big(-E_s^\top P_{\overline{\mathbf{m}}}^\top\mathbf{y} + \alpha_s^{-1}E_s^\top M\boldsymbol{\mu}_{s|t}(\mathbf{x}_t;\eta) + E_s^\top ME_s\boldsymbol{\mu}_{s|t}(\mathbf{x}_t;\eta)\big) + \mathcal{O}(\eta_s^4) .$$

with $M = P_{\overline{\mathbf{m}}}^\top P_{\overline{\mathbf{m}}}$, which is an orthogonal projection matrix since $M^\top = M$ and $P_{\overline{\mathbf{m}}}P_{\overline{\mathbf{m}}}^\top = I_{d_{\mathbf{y}}}$ and thus $M^2 = M$. We proceed by bouding each term of

$$-E_s^\top P_{\overline{\mathbf{m}}}^\top\mathbf{y} + \alpha_s^{-1}E_s^\top M\boldsymbol{\mu}_{s|t}(\mathbf{x}_t;\eta) + E_s^\top ME_s\boldsymbol{\mu}_{s|t}(\mathbf{x}_t;\eta)$$

separately. Define $\varepsilon_s := \|ME_s\|_{\mathrm{op}}$. Then, since $\mathbf{v} := P_{\overline{\mathbf{m}}}^\top \mathbf{y} \in \mathrm{range}(M)$, we have $M\mathbf{v} = \mathbf{v}$. Hence

$$E_s^\top P_{\overline{\mathbf{m}}}^\top y = E_s^\top M P_{\overline{\mathbf{m}}}^\top y = (ME_s)^\top (MP_{\overline{\mathbf{m}}}^\top \mathbf{y})$$

where we have used that $M^\top M = M$. By the operator norm inequality, and the fact that $\|P_{\overline{\mathbf{m}}}^\top \mathbf{y}\| = \|\mathbf{y}\|$, we get

$$\|E_s^\top P_{\overline{\mathbf{m}}}^\top \mathbf{y}\| \leq \|ME_s\|_{\mathrm{op}} \|MP_{\overline{\mathbf{m}}}^\top \mathbf{y}\| = \varepsilon_s \|P_{\overline{\mathbf{m}}}^\top \mathbf{y}\| = \varepsilon_s \|\mathbf{y}\|.$$

Next, using the same operator norm inequality we get that

$$\|E_s^\top M \boldsymbol{\mu}_{s|t}(\mathbf{x}_t; \eta)\| \leq \varepsilon_s \|M \boldsymbol{\mu}_{s|t}(\mathbf{x}_t; \eta)\|, \quad \|E_s^\top M E_s \boldsymbol{\mu}_{s|t}(\mathbf{x}_t; \eta)\| \leq \varepsilon_s^2 \|\boldsymbol{\mu}_{s|t}(\mathbf{x}_t; \eta)\|.$$

which yields the desired bound. $\qquad\square$

**Proposition 2** (Upperbound on $\varepsilon_s$). *We have that*

$$\varepsilon_s \leq \frac{\sigma_s^2}{\alpha_s} \frac{1}{\alpha_s^2 \lambda_{\min}(\Sigma) + \sigma_s^2}$$

*where $\lambda_{\min}(\Sigma)$ is the smallest eigenvalue of $\Sigma$.*

*Proof.* By noting that $(\alpha_s^2 \Sigma + \sigma_s^2 \mathrm{I})E_s = -\alpha_s^{-1}\sigma_s^2 \mathrm{I}_d$, we get the alternative expression

$$E_s = -\frac{\sigma_s^2}{\alpha_s}(\alpha_s^2 \Sigma + \sigma_s^2 \mathrm{I}_d)^{-1}.$$

By the submultiplicativity of the operator norm and the fact that $M$ is a non-trivial orthogonal projection matrix, we have that

$$\|E^\top M\|_{\mathrm{op}} \leq \|E\|_{\mathrm{op}} = \frac{\sigma_s^2}{\alpha_s} \frac{1}{\lambda_{\min}(\alpha_s^2 \Sigma + \sigma_s^2 \mathrm{I}_d)} \leq \frac{\sigma_s^2}{\alpha_s} \frac{1}{\alpha_s^2 \lambda_{\min}(\Sigma) + \sigma_s^2}.$$

$\qquad\square$

# B    DETAILS ABOUT THE EXPERIMENTS

## B.1    MODELS

We use both the SD 3 and SD 3.5 (medium) (Esser et al., 2024) models with the linear schedule $\alpha_t = 1 - t$ and $\sigma_t = t$. In all the experiments we run the zero-shot methods with a guidance scale of 2. The fine-tuned baseline, which we refer to as SD3 Inpaint, is based on the publicly available model[3] trained for inpainting with a ControlNet-augmented version of Stable Diffusion 3. It has been finetuned on a large dataset of approximately 12 million $1024 \times 1024$ image–mask pairs to directly predict high-quality inpainted completions conditioned on the masked image and the mask itself. We have found the model to perform well also on lower resolutions, despite not undergoing multi-resolution training. Examples of image editing of lower resolution images are presented in the the HuggingFace page of the smae project. We run this baseline using a guidance scale of 7 for optimal results.

Finally, all experiments use `bfloat16` for model forward passes (and backward passes for baselines that require it), with other computations performed in `float32`.

## B.2    MASK DOWNSAMPLING

To construct the mask in the latent space, we start from the original binary mask defined in pixel space. Since the encoder reduces spatial resolution by a fixed factor (here, 8), we downsample the pixel-space mask to match the resolution of the latent representation. This is done by applying bilinear interpolation with antialiasing. The resulting low-resolution mask captures the proportion of masked pixels within each latent receptive field. Finally, we threshold this downsampled mask at 0.95 to obtain a binary latent mask, slightly overestimating the masked region to prevent boundary artifacts during sampling.

---

[3] https://huggingface.co/alimama-creative/SD3-Controlnet-Inpainting

### B.3 IMPLEMENTATION OF THE BASELINES

Here, we give implementation details of the baselines. We stress that *each baseline is run in the latent space*, and thus no method computes the gradient w.r.t. the input of the decoder. We also manually tuned each baseline for the considered tasks. We provide the used hyperparameters in Table 8.

BLENDED-DIFF. We implemented Avrahami et al. (2023, Algorithm 1) following their official code[4]. The codebase includes an additional hyperparameter, `blending_percentage`, which determines at what fraction of the inference steps blending begins. We set it to zero, as applying blending across all steps produced the best results. A key detail is the original implementation is that the observed region (background) is re-noised to the noise level defined by the current timestep; see Avrahami et al. (2023, step 1-2 within the for loop in Algo 1), yet the reconstructed region (foreground) has less noise as it comes from applying a DDIM transition. This causes the background and foreground to follow different noise levels, and hence, introduces minor artifacts in the final reconstructions. We fixed this issue in our implementation by matching the two noise levels.

DAPS. We adapt Zhang et al. (2025, Algorithm 1) based on the released code[5] to the flow matching formulation. We found that using Langevin as MCMC sampler for enforcing data consistency works the best for low NFE regime.

DIFFPIR. We make Zhu et al. (2023, Algorithm 1) compatible with the flow matching formulation with step 4 being implemented in the case of mask operator. We found in practice that the hyperparameter $\lambda$ has little impact on the quality of reconstructions and hence we use the recommended values $\lambda = 1$[6]. On the other hand for the second hyperparameter $\zeta$, we find that using $\zeta = 0.3$ yielded the best reconstructions.

DDNM. We adapt the implementation in the released code[7] to the flow matching formulation with the step 4 in Wang et al. (2023b, Algorithm 3) being implemented for a mask operator. The official implementation uses a DDIM transition in step 5 of Algorithm 3 whose stochasticity is controlled by the hyperparemters $\eta$. As recommended, we set the latter to $\eta = 0.85$.

FLOWCHEF & FLOWDPS. For both algorithms, we adapt the implementations available in the released codes FLOWCHEF[8] [9] to our codebase. We observe that the two algorithms are quite similar, with FLOWDPS being distinct by adding stochasticity between iterations.

PNP-FLOW. We reimplement Martin et al. (2025, Algorithm 3) while taking as a reference the released code[10]. For the stepsizes on data fidelity term, we find that a constant scheduler with higher stepsize enables the algorithm to fit the observation, mitigate the smooth and blurring effects in the reconstruction and hence yield better reconstructions.

PSLD. We implement the PSLD algorithm provided in Rout et al. (2024b, Algorithm 2). We find that PSLD algorithm requires several diffusion steps, e.g. at least 150 diffusion steps, to yield good results. Unfortunately, we were not able to make it work well for the low NFE setup.

REDDIFF. We implement Mardani et al. (2024, Algorithm 1) based on the official code[11] and adapt it to the flow matching formulation. We initialize the algorithm with a sample for a standard Gaussian. For low NFE setups, we find that using a constant weight schedule yields better results, namely in terms fitting the observation and providing consistent reconstructions.

---

[4] https://github.com/omriav/blended-latent-diffusion
[5] https://github.com/zhangbingliang2019/DAPS
[6] https://github.com/yuanzhi-zhu/DiffPIR
[7] https://github.com/wyhuai/DDNM
[8] https://github.com/FlowDPS-Inverse/FlowDPS
[9] https://github.com/FlowChef/flowchef
[10] https://github.com/annegnx/PnP-Flow
[11] https://github.com/NVlabs/RED-diff

RESAMPLE. We reimplemented Song et al. (2024, Algorithm 1) based on the provided implementation details in Song et al. (2024, Appendix) and the reference code[12]. As noted in Janati et al. (2025a), we set the tolerance $\varepsilon$ for optimizing the data consistency to the noise level $\sigma_{\mathbf{y}}$. Since we are working with low NEFs, we set the frequency at which hard data consistency is applied (skip step size) to 5. That aside, we found that the algorithm requires several diffusion steps (200) in order to output good enough reconstructions. We note that removing the DPS step in the data consistency steps reduces the quality of the reconstructions.

Table 8: Hyperparameters for each algorithm (using the same notations as in their paper) and task variations. "—" indicates identical across tasks.

| Algorithm | $n_{\text{steps}}$ | Base hyperparameters | Latent tasks | | | | |
|---|---|---|---|---|---|---|---|
| | | | Half | Top | Bottom | Center | Strip |
| BLENDED-DIFF | 50 | `blending_percentage = 0` | — | — | — | — | — |
| DAPS | 50 | $N_{\text{ode}} = 2$ 
 `MCMC steps = 20` 
 $\beta_y = 10^{-2}$ 
 `Min ratio = 0.43` 
 `MCMC sampler = Langevin` 
 $\rho = 1$ | $\eta_0 = 2 \times 10^{-5}$ | $\eta_0 = 3 \times 10^{-5}$ | $\eta_0 = 2 \times 10^{-5}$ | $\eta_0 = 9 \times 10^{-6}$ | $\eta_0 = 2 \times 10^{-5}$ |
| DDNM | 50 | $\eta = 0.85$ | — | — | — | — | — |
| DIFFPIR | 50 | $\lambda = 1$ 
 $\zeta = 0.3$ | — | — | — | — | — |
| FLOWCHEF | 50 | `step size = 0.9` 
 `grad_descent_steps = 10` | — | — | — | — | — |
| FLOWDPS | 50 | `grad_descent_steps = 3` | step_size = 20 | step_size = 10 | step_size = 10 | step_size = 10 | step_size = 10 |
| PNP-FLOW | 50 | $\alpha = 1.0$ 
 `lr style = constant` | $\gamma_n = 0.8$ | $\gamma_n = 1.3$ | $\gamma_n = 1.4$ | $\gamma_n = 0.8$ | $\gamma_n = 0.8$ |
| PSLD | 50 | `DDIM_param = 1.0` | $\gamma = 0.01$ 
 $\eta = 0.01$ | $\gamma = 0.01$ 
 $\eta = 0.01$ | $\gamma = 0.01$ 
 $\eta = 0.01$ | $\gamma = 0.05$ 
 $\eta = 0.1$ | $\gamma = 0.1$ 
 $\eta = 0.5$ |
| REDDIFF | 50 | `lr = 0.2` 
 `grad_term_weight = 0.25` 
 `obs_weight = 1.0` | — | — | — | — | — |
| RESAMPLE | 50 | $C = 5$ 
 `grad_descent_steps = 200` 
 $\gamma_{\text{scale}} = 40.0$ 
 $\text{lr}_{\text{pixel}} = 10^{-2}$ 
 $\text{lr}_{\text{latent}} = 5 \times 10^{-3}$ | — | — | — | — | — |
| DING (ours) | 25 | $\eta = \sigma_s(1 - \alpha_s)$ | — | — | — | — | — |

## C  EXAMPLES OF RECONSTRUCTIONS

Here, we provide a side-by-side comparison of the DING and the considered baselines on image editing tasks via inpainting on `PIE-Bench`. The red semi-transparent layer in the first column shows the masked region to be edited and the text in the left-hand side of each row represents the editing prompt.

In the follow work Ghorbel et al. (2026), we provide extended experiments that include other datasets, models, as well as the editing task on videos.

*(See the next pages for the gallery of examples)*

---

[12]https://github.com/soominkwon/resample

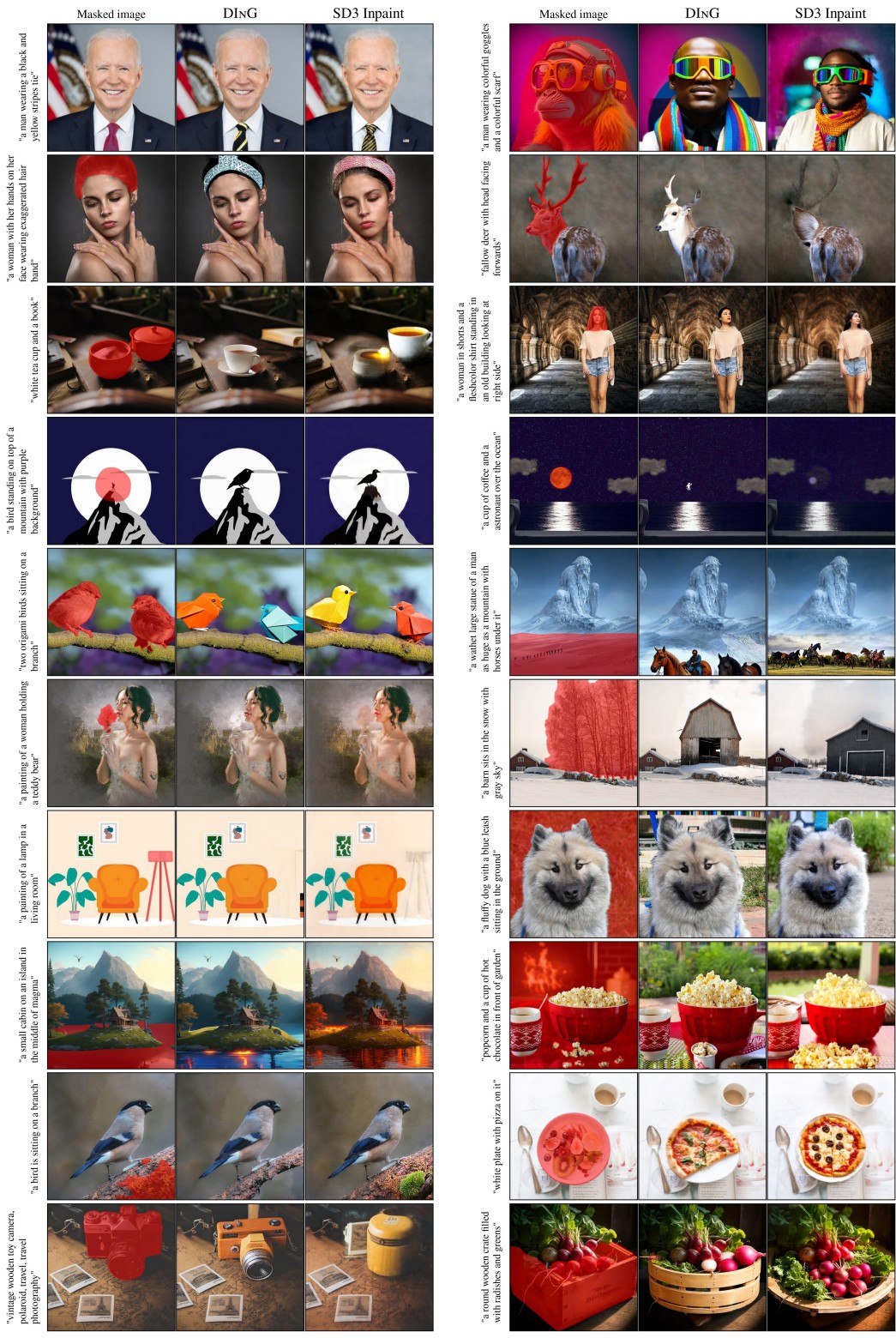

Figure 7: Comparison of DIɴG and finetuned SD3 on `PIE-Bench`. Both methods have the same runtime of 2.2s.

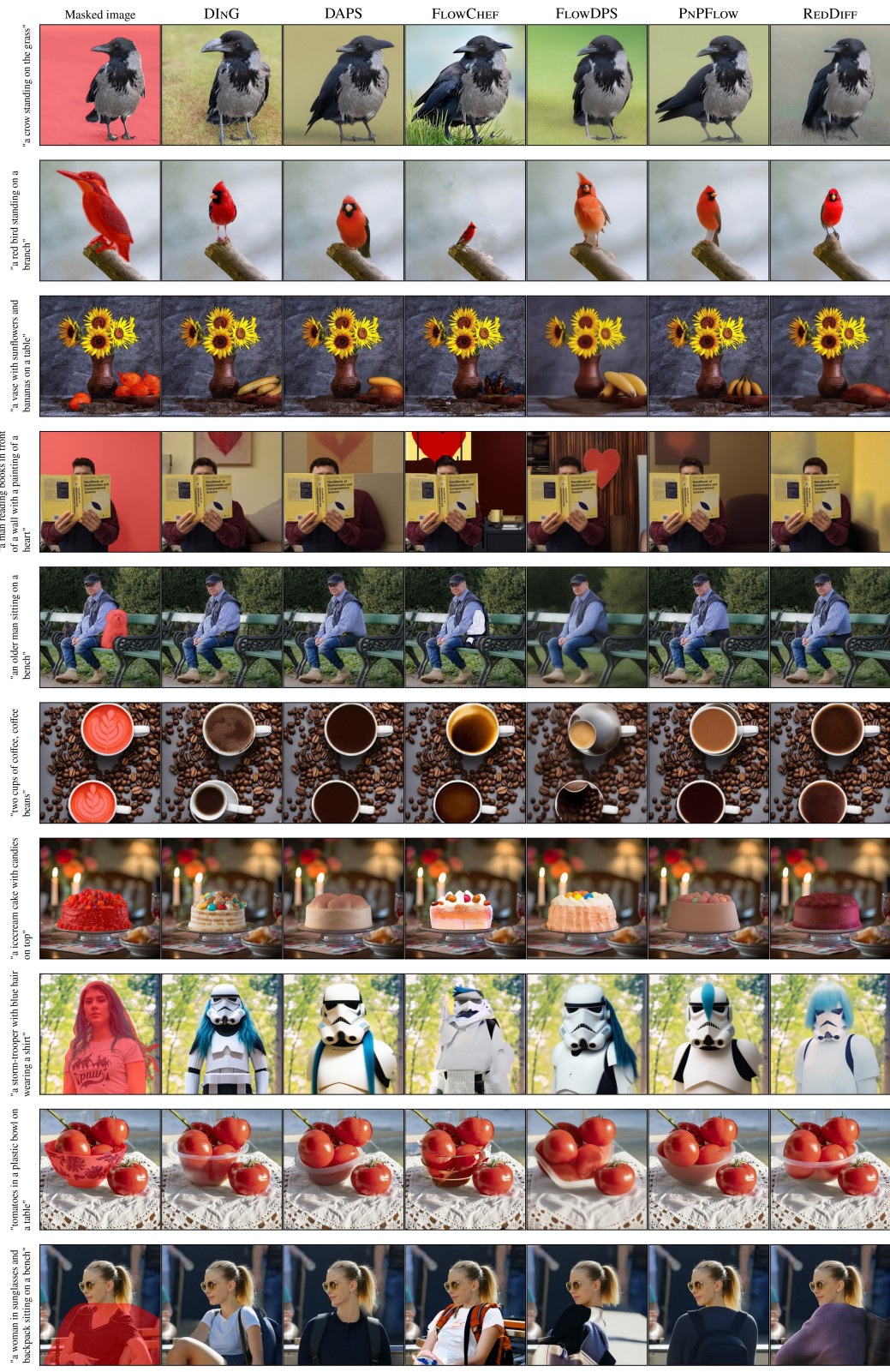

Figure 8: Comparison of DING and zero-shot baselines on `PIE-Bench`. All methods use 50 NFEs.

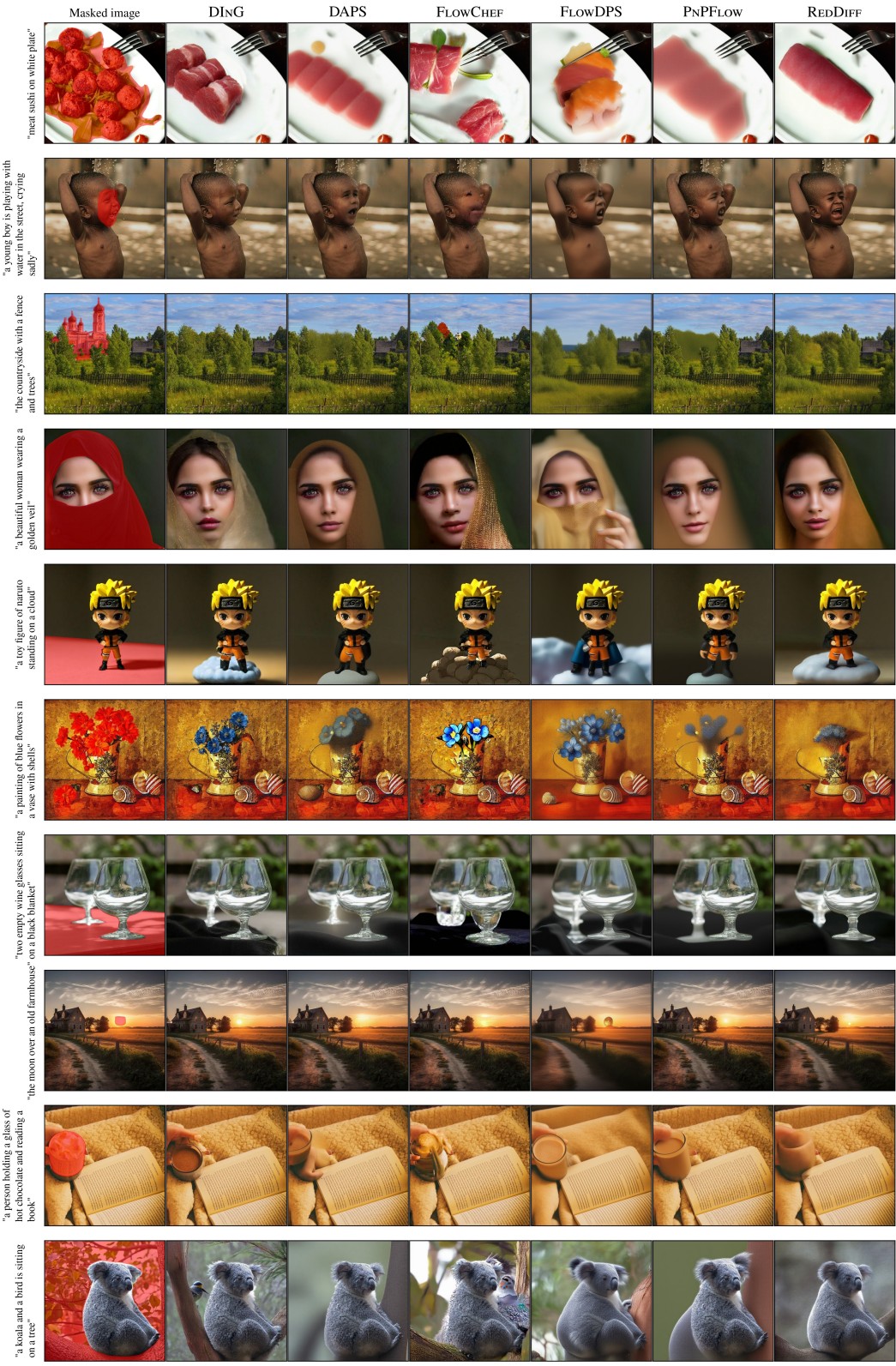

Figure 9: Comparison of DING and zero-shot baselines on `PIE-Bench`. All methods use 50 NFEs.

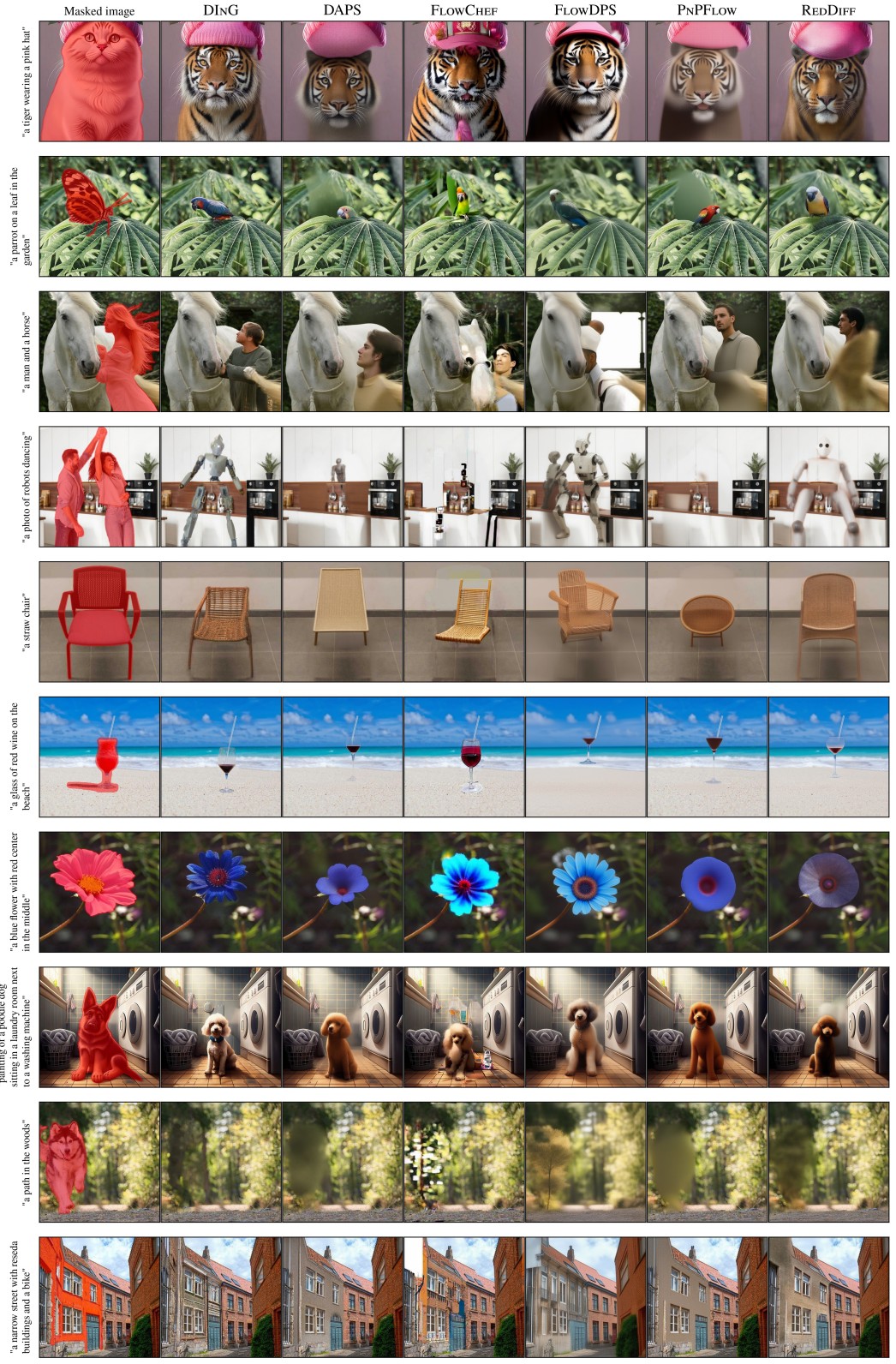

Figure 10: Comparison of DING and zero-shot baselines on `PIE-Bench`. All methods use 50 NFEs.

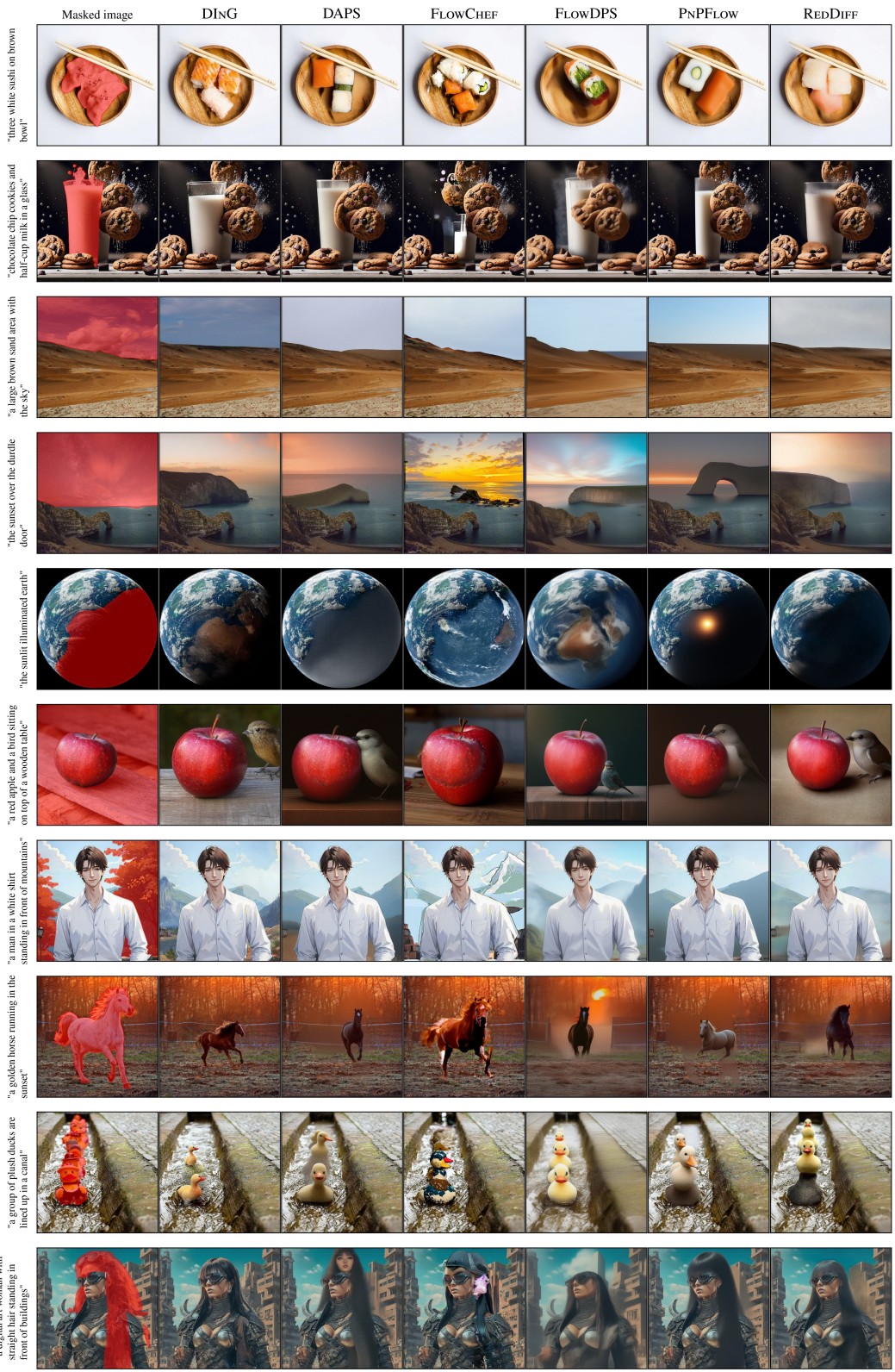

Figure 11: Comparison of DING and zero-shot baselines on `PIE-Bench`. All methods use 50 NFEs.

