# OpenReview forum: "Efficient Zero-shot Inpainting with Decoupled Diffusion Guidance"
_ICLR.cc/2026/Conference — ICLR 2026 Poster_

### Official Review · Reviewer_S6KU · 2025-10-31

**Soundness:** 3
**Presentation:** 3
**Contribution:** 2
**Rating:** 6
**Confidence:** 4

**Summary:**

The paper proposes DING, a zero-shot diffusion-based inpainting framework that eliminates vector-Jacobian product (VJP) computations by decoupling the likelihood evaluation from the denoiser input. The method claims efficiency gains (reduced memory/runtime) while maintaining or surpassing the quality of prior zero-shot and fine-tuned baselines on FFHQ, DIV2K, and PIE-Bench datasets.

**Strengths:**

1. The paper is well-written, and the mathematical exposition is clear.
2. The authors provide extensive experimental results across multiple datasets and metrics, consistently showing DING’s advantages. They also make a commendable effort to ensure fair comparisons, clearly documenting baseline hyperparameters.
3. The inclusion of runtime and memory analyses further strengthens the empirical evaluation.

**Weaknesses:**

1. The mathematical derivation in Section 3.2 relies on several unverified approximations (see questions below). These steps seem heuristic and lack theoretical justification. Moreover, the closed-form update in Eq. (3.4) depends critically on Gaussian-Gaussian conjugacy, which holds only for linear masking operators. As a result, the proposed method is mathematically fragile and inherently task-specific. Given this restriction, there is little reason to prefer a task-agnostic zero-shot formulation over simply training a dedicated inpainting model, since DING offers no clear theoretical or practical advantage beyond the inpainting setting.
2. The paper evaluates against general posterior-sampling methods but omits comparison to established fast zero-shot inverse-problem solvers such as DDNM and DiffPIR, which also target inpainting and avoid VJP overhead. DDNM, in particular, is well known for producing high-quality inpainting results when $σ_y$=0.
3. Although the derivation includes a measurement-noise parameter ($σ_y$) in the likelihood and posterior update (Eq. 3.4), all experiments appear to assume a noise-free setting ($σ_y$=0.0). Consequently, the practical behavior of DING under noisy observations is unknown. Since $σ_y$ directly controls the trade-off between data fidelity and prior regularization, omitting this analysis limits the method’s generality and makes it unclear whether the surrogate posterior remains effective when measurements are noisy.
4. DING is reported to outperform an SD3.5-Inpainting model trained for 12 M iterations with ControlNet conditioning. This is counterintuitive to me, as I’d expect a model of that scale, trained so extensively for a dedicated inpainting task, to possess a stronger conditional prior in principle. Although the authors match wall-clock runtime between DING and the fine-tuned SD3.5-Inpainting model, the comparison remains asymmetric. DING performs 56 denoiser evaluations (iterative posterior refinement), whereas the fine-tuned model runs only 28 steps of one-shot generation.

**Minor Comments:**
- Lines 397-402, the authors repeat themselves across two consecutive sentences.
- Including results on random inpainting masks would also strengthen the evaluation and demonstrate robustness across mask types.

**Questions:**

1. *Regarding weakness 1:* In equation (3.4), authors define the mixture of Gaussians (over $\mathbf{z}_s$) that DING samples from. But the true posterior transition π($\mathbf{x}_s​$∣$\mathbf{x}_t$​, $\mathbf{y}$) is not Gaussian as it depends nonlinearly on the denoiser. So under what conditions does the Gaussian mixture in Eq. (3.4) approximate the true posterior transition? Is there a theoretical approximation guarantee?
2. *Regarding weakness 1:* By decoupling the denoiser input from the latent, does DING still produce unbiased posterior means and variances?
3. *Regarding weakness 3:* Can authors comment on DING’s performance when $σ_y$ > 0? Does the method remain stable and accurate when the measurement noise increases?
4. *Regarding weakness 4:* Would DING still outperform the fine-tuned inpainting model if both used the same number of diffusion steps, and the fine-tuned model employed its default CFG weighting of 7.0 instead of the reduced value of 4.0?
5. *Regarding weakness 4:* In Figure 4 and Table 4, the fine-tuned inpainting model appears to modify observed pixels, resulting in lower cPSNR. This behavior is unexpected. Can the authors explain the cause? Is it due to suboptimal hyperparameters, the 512×512 test resolution differing from the model’s 1024×1024 training resolution, or an architectural limitation?

---

> ### Author Response · Authors · 2025-11-20
>
> Dear Reviewer,
>
> We thank the reviewer for the detailed feedback. Below we address the remaining concerns.
>
> > DING is reported to outperform an SD3.5-Inpainting model trained for 12 M iterations with ControlNet conditioning. This is counterintuitive to me, as I’d expect a model of that scale, trained so extensively for a dedicated inpainting task, to possess a stronger conditional prior in principle. Although the authors match wall-clock runtime between DING and the fine-tuned SD3.5-Inpainting model, the comparison remains asymmetric. DING performs 56 denoiser evaluations (iterative posterior refinement), whereas the fine-tuned model runs only 28 steps of one-shot generation.
>
> While this may seem surprising, we would like to point to the reviewer that even on the huggingface page of the SD3 Inpainting model that we use, there are displayed examples where there is a lack of consistency with observed part of the image; see for example the third row here: https://huggingface.co/alimama-creative/SD3-Controlnet-Inpainting/blob/main/images/5_compressed.png
>
> where it can be seen that the reconstructed outputs visibly alter context adjacent to the masked region (third panel sequence). This behavior is also consistent with the Diffusers documentation, which notes that inpainting pipelines “generally change the unmasked parts of an image” to smooth the transition, and that forcing the unmasked area to remain identical can introduce unnatural seams; see the section “Preserve unmasked areas” [1]. We emphasize that we reproduced all competitors with care and aimed for unbiased comparisons (matching wall-clock runtime and reporting actual memory/time, not nominal NFEs). For SD3-Inpainting, the ControlNet incurs a substantial per-step overhead, so equal NFE does not imply equal runtime; accordingly, our main table uses fewer SD3 Inpainting steps to match runtime. For completeness, we also reran SD3-Inpainting with the same NFE count as our method; our approach still achieves the best overall trade-off (fidelity, perceptual quality, and consistency) in that setting as well. We updated the same table in the main paper, please see Section 4.1.
>
> [1] https://huggingface.co/docs/diffusers/v0.22.3/en/using-diffusers/inpaint
>
> > Would DING still outperform the fine-tuned inpainting model if both used the same number of diffusion steps, and the fine-tuned model employed its default CFG weighting of 7.0 instead of the reduced value of 4.0?
>
> Here we would like to report that our paper contains a typo and that we actually used a CFG weight of 7 and not 4. We are sorry for this. This can be verified in the codebase provided with the algorithm at configs/sampler/alimama_controlnet.yaml, line 5.
>
> > The mathematical derivation in Section 3.2 relies on several unverified approximations (see questions below). These steps seem heuristic and lack theoretical justification. Moreover, the closed-form update in Eq. (3.4) depends critically on Gaussian-Gaussian conjugacy, which holds only for linear masking operators. As a result, the proposed method is mathematically fragile and inherently task-specific.
>
> We thank the reviewer for this comment and agree that the derivation involves approximations. Our goal, however, is to introduce a practical and verifiable proof of concept showing that such decoupled surrogates can approximate DPS-like transitions efficiently. The Gaussian conjugacy in Eq. (3.4) is indeed exact under the linear inpainting operator, which is the problem we set out to tackle in this paper.
>
> Following your comment we now provide a sanity-check analysis in Appendix A.5 demonstrating that, in the Gaussian case, the approximation error with respect to the DPS transition is provably controlled. This supports the soundness of the formulation in its intended regime and establishes a principled foundation for exploring extensions beyond linear masking operators in future work.
> Regarding the concern about task specificity, we note that this limitation is inherent to all zero-shot methods operating in the latent space without backpropagating through the decoder. Once we fix this constraint to ensure fast inference, only the inpainting operator can be safely and directly lifted to the latent domain, as it remains linear and compatible with the Gaussian conjugacy used in our derivation. Other operators, such as those for deblurring or super-resolution, require explicit modeling in pixel space or additional training to define a corresponding latent operator. Hence, rather than being a limitation unique to our approach, this reflects a general property of efficient, training-free latent diffusion methods.

---

> > ### Author Response · Authors · 2025-11-20
> >
> > > Given this restriction, there is little reason to prefer a task-agnostic zero-shot formulation over simply training a dedicated inpainting model, since DING offers no clear theoretical or practical advantage beyond the inpainting setting.
> >
> > We respectfully disagree with the reviewer’s assessment. Inpainting is one of the most common and practically relevant applications of diffusion models, yet fine-tuning large conditional models such as SD3-Inpainting requires significant computational resources and large, curated datasets that most users do not have access to. In practice, users often need to wait for model providers to release specialized inpainting checkpoints, which can take considerable time, as seen with SD3. In contrast, DING operates directly on any pretrained diffusion prior without retraining or labeled data. This makes it immediately usable for new foundation models and adaptable to domains where fine-tuned versions are not available. In addition, our experiments show that DING not only avoids the high training cost but also outperforms the fine-tuned SD3-Inpainting model under matched runtime, providing both practical and empirical advantages within the inpainting setting.
> >
> > > The paper evaluates against general posterior-sampling methods but omits comparison to established fast zero-shot inverse-problem solvers such as DDNM and DiffPIR, which also target inpainting and avoid VJP overhead. DDNM, in particular, is well known for producing high-quality inpainting results when $\sigma_y =0$.
> >
> > Thank you for this suggestion. We have now included a comparison to DDNM and DiffPIR under the same runtime budget; see the updated experimental section in the paper. These baselines are indeed strong, as they have a slight edge over our method in terms of cPSNR.  Nevertheless, our method continues to achieve the best overall trade-off across fidelity, perceptual, and efficiency metrics. We have also added a new paragraph in Section A.2 in the paper to explain the main difference with DiffPIR and DDNM. Finally, following the recommendation of another reviewer, we have included the baseline blended latent diffusion [1].
> >
> > > Although the derivation includes a measurement-noise parameter ($\sigma_y$) in the likelihood and posterior update (Eq. 3.4), all experiments appear to assume a noise-free setting ($\sigma_y=0.0$). Consequently, the practical behavior of DING under noisy observations is unknown. Since directly controls the trade-off between data fidelity and prior regularization, omitting this analysis limits the method’s generality and makes it unclear whether the surrogate posterior remains effective when measurements are noisy.
> >
> > We thank the reviewer for pointing this out. As stated in the Experiments section, we use a nonzero noise level of $\sigma_y = 0.01$ across all benchmarks; thus, all methods are evaluated under a mildly noisy setting. We would like to highlight here that one limitation of lifting the inpainting operator to the latent space is that the encoder may struggle to encode properly the noisy observation, and that the Gaussian likelihood model in the latent space may no longer be adapted to the latent observation. This is a limitation of performing image editing in the latent space and applies to all the baselines considered. Nonetheless, we believe this limitation is not fundamental. In practice, one can design a pipeline where the noisy image is first denoised in pixel space or by a separate latent diffusion model, and the resulting clean latent representation is then used as input for DING together with the mask. This would effectively restore the Gaussian observation model assumed in our formulation while preserving the efficiency of latent-space inference.
> >
> > > Including results on random inpainting masks would also strengthen the evaluation and demonstrate robustness across mask types.
> >
> > Thank you for the suggestion, we will include this setting in the final version of the paper.
> >
> > **Further paper updates**: On top of the modifications mentioned in our rebuttal, we have also included a Limitation section in the appendix highlighting when our method fails. See Section A.4. All of the modifications are highlighted in green in the updated PDF on openreview.
> >
> > We again thank the reviewer for their constructive feedback and valuable suggestions. We remain open to any further questions or suggestions the reviewer may have.
> >
> > [1] Avrahami, O., Fried, O. and Lischinski, D., 2023. Blended latent diffusion.

---

### Official Review · Reviewer_PHGS · 2025-11-01

**Soundness:** 3
**Presentation:** 3
**Contribution:** 3
**Rating:** 6
**Confidence:** 4

**Summary:**

This paper introduces DING (Decoupled INpainting Guidance), a zero-shot diffusion-based inpainting method that eliminates the need for backpropagation through the denoiser during inference. Existing zero-shot inpainting methods require computing vector–Jacobian products (VJPs) through the diffusion denoiser network at every reverse step, which leads to high computational and memory costs.
DING addresses this by proposing a decoupled likelihood approximation, where the denoiser is evaluated at an independent proxy variable rather than the current latent state. This removes the dependency between the denoiser and the transition density, enabling closed-form Gaussian posterior sampling without backpropagation. The authors evaluate DING on multiple benchmarks using Stable Diffusion 3.5 as the pretrained prior. DING demonstrates superior trade-offs between reconstruction fidelity, perceptual realism, and computational efficiency compared to recent zero-shot inpainting and posterior-sampling methods such as FLOWCHEF, DAPS, and PSLD. The proposed method even outperforms a fine-tuned SD3 inpainting model despite requiring no additional training.

**Strengths:**

+ The computational inefficiency of current zero-shot diffusion-based inpainting methods is a relevant and well-identified issue.
+ The “decoupling” of the denoiser input and the posterior likelihood evaluation is a simple but effective modification that removes the need for costly gradient computations.
+ Experimental results show clear improvements in runtime and memory consumption.
+ DING achieves consistently better or comparable scores in FID, pFID, LPIPS, and cPSNR across datasets and even outperforms fine-tuned baselines under a fair computational budget.

**Weaknesses:**

- While the decoupling trick is brilliant, the method builds upon a well-studied line of “posterior sampling with diffusion priors”, such as DPS, FLOWDPS, DAPS, PnP-FLOW, and mainly contributes an engineering-level efficiency improvement rather than a new theoretical insight.
- The derivation of the closed-form Gaussian posterior is mathematically sound, but the paper lacks deeper analysis or guarantees about the approximation error introduced by the decoupling. The Bayesian justification is somewhat heuristic.
- DING is designed only for inpainting, as the Gaussian observation model is directly tied to pixel-wise missing-region reconstruction. The paper’s claim that the approach can generalize to “other inverse problems” remains speculative.
- The paper is dense with mathematical notation and references, for example, multiple layers of priors, posterior transitions, likelihood surrogates, which makes it unnecessarily hard to follow, especially in Sections 2–3.
- All experiments use Stable Diffusion 3.5 with standard image domains. It would strengthen the work to include more challenging or domain-specific cases (e.g., medical or scientific imaging) where zero-shot efficiency matters most.

**Questions:**

1. Can the authors clarify how runtime and memory are measured? Specifically, does the reported 2.9s runtime include both proxy sampling and denoiser evaluations (2× per step)?
2. Since the derivation assumes a Gaussian likelihood over masked pixels, how easily can this framework extend to deblurring, super-resolution, or compressive sensing tasks?

---

> ### Author Response · Authors · 2025-11-20
>
> Dear Reviewer,
> We thank the reviewer for their detailed feedback. Below we address your remaining concerns.
>
> > Can the authors clarify how runtime and memory are measured? Specifically, does the reported 2.9s runtime include both proxy sampling and denoiser evaluations (2× per step)?
>
> Yes, the reported runtime includes both proxy sampling and denoiser evaluations. To ensure a fair comparison, we use 25 diffusion steps (each requiring two denoiser calls) to match 50 NFEs. We also note that Reviewer S6KU requested an additional comparison against the fine-tuned baseline using a larger NFE budget. Despite its higher runtime due to the ControlNet overhead, our method still outperforms it.
>
> > While the decoupling trick is brilliant, the method builds upon a well-studied line of “posterior sampling with diffusion priors”, such as DPS, FLOWDPS, DAPS, PnP-FLOW, and mainly contributes an engineering-level efficiency improvement rather than a new theoretical insight.
>
> We respectfully disagree with this assessment, as our approach is fundamentally different from prior works. To the best of our knowledge, the proposed decoupling trick, i.e. evaluating the denoiser at an independent proxy rather than the current latent and the parameterization used, has not been explored in the literature. This design yields consistent strong empirical gains, offering a new perspective that we believe can further inspire future works. Moreover, we now provide a theoretical sanity check in Appendix A.5, where we derive an exact analysis in the Gaussian setting, showing that the approximation error with respect to the DPS transition remains controlled. This result supports the soundness of the proposed approximation and distinguishes our contribution from purely engineering-level improvements.
>
> > DING is designed only for inpainting, as the Gaussian observation model is directly tied to pixel-wise missing-region reconstruction. The paper’s claim that the approach can generalize to “other inverse problems” remains speculative.
>
> We appreciate the reviewer’s observation and would like to clarify that our paper **does not** claim general applicability beyond inpainting. As stated in the Limitations section, DING is specifically designed for inpainting, where the observation operator can be naturally lifted to the latent space. We mention broader inverse problems only as a potential research direction, not as an asserted capability of the current method. The main challenge in extending DING lies in the treatment of the forward operator. For inpainting, the mask translates directly into a simple linear operator in the latent domain, which underpins DING’s efficiency and accuracy. In contrast, for tasks such as super-resolution or deblurring, this mapping is not straightforward—there is no direct way to express the operator in latent space without training a dedicated latent operator (as explored, for example, in [1]). Extending our framework in a training-free manner would therefore require operating in pixel space and differentiating through the decoder. In this setting, our likelihood surrogate can still be applied by approximating the conditional transition on $z_t$ (as described before Eq. 3.3) via Gaussian variational inference, but we found this approach sensitive to the parameterization of the Gaussian approximation and prone to overly smooth reconstructions if not carefully designed. Addressing these challenges to enable stable extensions beyond inpainting is part of our ongoing work.
>
> > The paper is dense with mathematical notation and references, for example, multiple layers of priors, posterior transitions, likelihood surrogates, which makes it unnecessarily hard to follow, especially in Sections 2–3.
>
> We thank the reviewer for their suggestion and are open to improving the presentation of our manuscript. However, we respectfully disagree with the characterization that there are “multiple layers” of prior, posterior, and likelihood surrogates. As currently written, the paper introduces only two types of such objects: the ideal quantities and their approximations. As a result, we find it difficult to identify which notations could be removed without undermining the clarity or internal consistency of the methodology and its theoretical justification. We would therefore be grateful if the reviewer could provide more specific guidance or examples to help us address this concern.

---

> > ### Author Response · Authors · 2025-11-20
> >
> > **Further paper updates**: Following suggestions from other reviewers, we have incorporated three additional baselines, DDNM, DiffPIR, and Blended Diffusion, and our method continues to achieve the best trade-off across quality, runtime, and memory. These new results are provided in the Section 4.1 in the main paper. Furthermore, we added explicit transitions comparisons against DiffPIR and DDNM in Appendix A.2, and introduced a new section discussing the limitations of our method (Section A.4). All of the modifications are highlighted in green in the updated PDF on openreview.
> >
> > We again thank the reviewer for their constructive feedback and valuable suggestions. We remain open to any further questions or suggestions the reviewer may have.
> >
> > [1] Raphaeli, R., Man, S. and Elad, M., 2025. Silo: Solving inverse problems with latent operators.

---

### Official Review · Reviewer_MmE3 · 2025-11-01

**Soundness:** 3
**Presentation:** 2
**Contribution:** 3
**Rating:** 6
**Confidence:** 4

**Summary:**

This paper proposes a new idea for diffusion-based inpainting based on classifier guidance. Classifier guidance is a well-established technique for conditioning, but it has some shortcomings (the $t$-step conditioning distributions have to be all known). The paper expands on some well-known approximation ideas to devise a new form of transition, which does not require the additional backpropagation step for the classifier guidance. Based on the approximated conditioned transition formulation, the paper introduces an auxiliary variable to make the transition probability a mixture distribution. Now, the transition can be performed exactly in a Monte Carlo manner. Experiments demonstrate that the proposed method achieves superb performance with large gaps.

**Strengths:**

- Clever idea. I like the reformulation technique. Incorporating the Dirac approximation idea into the conditional transition formulation is one thing, but the auxiliary variable idea is also a nice touch.

- State-of-the-art performance with large gaps.

**Weaknesses:**

- I'm generally convinced by the formulation and the results, but one question remains: Seeing "Related methods" section and Appendix A.2, I cannot help but think whether the proposed method is just providing an alternative story to already existing methods. Appendix A.2 highlights some technical differences, but it is difficult to grasp the distinct value of the proposed method. What is the practical/core advantage of the proposed method in terms of theory/effectiveness compared to the methods mentioned in those sections? On the other hand, PnPFlow shows much worse performance in the experiments. Why is this?

- My review score is currently 6, but it is solely due to the above concern. I'll decide my final score after seeing the rebuttal.

**Questions:**

Please see the above weaknesses.

---

> ### Author Response · Authors · 2025-11-20
>
> Dear Reviewer,
> We would like to thank you for your feedback on our paper. Below we address your remaining concerns.
>
> > I'm generally convinced by the formulation and the results, but one question remains: Seeing "Related methods" section and Appendix A.2, I cannot help but think whether the proposed method is just providing an alternative story to already existing methods. [...]
>
> While preparing the paper, we explicitly derived the update rules of related methods to ensure our approach is not a re-statement of an existing algorithm. In Appendix A.2, we present side-by-side updates for several baselines (including MCGDIFF) and show the clear distinctions with our method. We have also expanded this section to clarify the precise differences from additional methods that can appear superficially similar, such as DiffPIR and DDNM. We are thus confident that our method is not a re-statement of existing methods.
>
> > it is difficult to grasp the distinct value of the proposed method. What is the practical/core advantage of the proposed method in terms of theory/effectiveness compared to the methods mentioned in those sections?
>
> The main distinctive value of our method is its simplicity and efficiency: it provides a practical, VJP-free approximation of the DPS likelihood that preserves reconstruction quality while significantly reducing runtime and memory cost. The goal of this work is to demonstrate that such decoupled approximations can indeed perform competitively in practice, serving as a proof of concept rather than a fully developed theoretical framework. Although a complete analysis is challenging, we have derived sanity checks in the Gaussian setting (Appendix A.5) showing that the approximation error with respect to DPS remains controlled. This supports the idea that our approach offers a sound and computationally lightweight alternative to exact DPS guidance.
>
> > On the other hand, PnPFlow shows much worse performance in the experiments. Why is this?
>
> Regarding the performance of PnP-Flow, we note that similar behavior is already evident in the original paper, where all inpainting results exhibit blurred and low-frequency reconstructions within the masked regions (see Fig. 7 and the bottom rows of Figs. 9–12). This is fully consistent with our own observations (see Fig. 2 and additional examples in the appendix), where PnP-Flow fails to recover sharp or semantically coherent details. We emphasize that our implementation strictly follows the official release and we tuned the parameters for SD3 so as to avoid as much as possible the blurry reconstructions; see the PnP-Flow section in Section B.3. Importantly, the PnP-Flow results in the original work were obtained using relatively small pretrained models, not large-scale backbones such as Stable Diffusion 3. Consequently, its performance in this regime remains unexplored, and our results provide the first quantitative and qualitative assessment of this setting.
>
> **Further paper updates**: Following suggestions from other reviewers, we have incorporated three additional baselines, DDNM, DiffPIR, and Blended Diffusion, and our method continues to achieve the best trade-off across quality, runtime, and memory. We also extended the comparison with the SD3 inpainting baseline under higher NFE budgets, where our approach still outperforms it. These new results are provided in the Section 4.1 in the main paper. Furthermore, we added explicit transitions comparisons against DiffPIR and DDNM in Appendix A.2, and introduced a new section discussing the limitations of our method (Section A.4). All of the modifications are highlighted in green in the updated PDF on openreview.
> We again thank the reviewer for their constructive feedback and valuable suggestions. We remain open to any further questions or suggestions the reviewer may have.

---

> > ### Comment · Reviewer_MmE3 · 2025-11-20
> >
> > Although I appreciate the added details in those sections, I'm still not entirely convinced by the explanations. What they basically show is the side-by-side comparisons of detailed transition probabilities. I do appreciate this and can see that they are indeed different, but providing a deeper insight into these differences (rather than just comparing the detailed expressions) is missing. I didn't read the other papers (of the methods shown in those sections), so it is difficult to grasp the core conceptual differences (which eventually produce those detailed differences).
> >
> > In other words, currently, it is difficult to judge whether those detailed differences are merely something like the 'simpler' differences between VE and VP methods in diffusion models, or whether they are coming from more fundamental differences. Please provide a deeper discussion of the fundamental (or conceptual) differences between those methods.

---

> > > ### Author Response · Authors · 2025-11-20
> > >
> > > We thank the reviewer for their rapid feedback. We understand that the reviewer is seeking a more qualitative explanation of the difference between our algorithm and the other baselines that may explain the performance improvement.
> > >
> > > For the inpainting problem considered in the paper, note that because of the forward process we expect that marginally, $X_s[\overline{m}]$ (the unmasked part of the noised state) to be roughly distributed according to $\mathcal{N}(\alpha_s y, \sigma^2 \_s I \_{d\_y})$. This shows that during the reverse process, we should constrain $X_s[\overline{m}]$ sampled given $X_t$ to "look like" a sample from this distribution. One naive way to do this is to simply use a sample $Y_s = \alpha_s y + \sigma \_s Z$ with $Z$ a standard Gaussian noise, and then use as likelihood $N(Y_s; x_s[\overline{m}], \sigma^2 \_* I \_{d \_y})$ where $\sigma_*$ is a hyperparameter to tune. This works poorly in practice due to the high sensitivity to the sampled noise $Z$ and the hyperparameter $\sigma_*$.
> > >
> > > The prior works such as MCGDIFF or PnP-Flow implement a similar idea by relying on a rather mis-specified likelihood model. In equation 4.1 in the revised paper we write that MCGDIFF basically uses $N(\alpha \_s y; x \_s[\overline{m}], \sigma^2 \_{s} I \_{d\_y})$ (assuming $\tau=0$) as likelihood for the approximate posterior transition. This likelihood model basically assumes that $\alpha_s y$, **which has no noise**, is an observation of $\alpha \_s X_s[\overline{m}] + \sigma^2 _s Z$, which has a significant amount noise. Due to this variance mismatch, we consider this to be a mis-specified model.
> > >
> > > To circumvent these issues, we draw inspiration from the DPS approximation 2.8 in the paper. The mild approximation we introduce in equation 3.2 states, after shuffling the various terms, that the likelihood should as observation $\alpha_s y + \sigma_s \hat{x}^\theta _1(z_s)[\overline{m}]$ instead; see Section A.1. This is very similar to what we mention in the first paragraph, with the exception that the standard Gaussian sample is replaced by the noise prediction $\hat{x}^\theta _1(z_s)[\overline{m}]$ (which approximates the expectation of the Gaussian noise given $z_s$). This resolves the variance mismatch, defines the appropriate standard deviation in the likelihood, and enables a more principled blending of information from the current state and the observation.
> > >
> > > We hope that this clarifies the qualitative differences with these prior works.

---

### Official Review · Reviewer_SWvq · 2025-11-02

**Soundness:** 3
**Presentation:** 3
**Contribution:** 3
**Rating:** 6
**Confidence:** 3

**Summary:**

This paper introduces DING (Decoupled INpainting GuidANCE), a zero-shot inpainting method that leverages a novel likelihood surrogate to enable efficient sampling with pre-trained diffusion models. By decoupling the denoiser evaluation from the likelihood surrogate, DING eliminates the need for costly vector-Jacobian products (VJPs) or backpropagation through the denoiser at each sampling step, leading to significant reductions in runtime and memory usage. Extensive experiments on benchmarks including FFHQ, DIV2K, and PIE-Bench demonstrate that DING achieves a strong trade-off between fidelity and realism, outperforming both state-of-the-art zero-shot methods and a fine-tuned Stable Diffusion 3 inpainting model, particularly under low NFE (number of function evaluations) budgets.

**Strengths:**

1. The proposed decoupled likelihood surrogate enables VJP-free sampling, directly addressing a major bottleneck in prior zero-shot diffusion guidance methods. This results in significant reductions in both runtime and memory usage, as empirically validated in Table 1.
2.  Across three challenging inpainting benchmarks (FFHQ, DIV2K, PIE-Bench), DING consistently outperforms existing zero-shot baselines and even surpasses a specialized fine-tuned SD3 model. The results are supported by both quantitative metrics (Tables 2–7) and compelling qualitative examples (Figures 4–8).
3.   The method operates efficiently in latent space and requires only forward passes through the denoiser, making it highly suitable for real-world applications with constrained computational resources.
4. The paper includes thorough ablation studies (e.g., on the necessity of doubled NFEs per step in Table 6, and DDIM noise schedules in Table 7), and baselines are carefully tuned to ensure fair comparisons.

**Weaknesses:**

1. While the decoupled likelihood surrogate is a clear improvement in efficiency and simplicity, it can be viewed as an incremental extension of existing VJP-free or mixture-based guidance methods (e.g., Wu et al., 2023; Janati et al., 2024/2025a). The paper could more explicitly articulate its conceptual departure from these prior works to better highlight its foundational contribution.
 2. The manuscript lacks a systematic analysis of failure cases or challenging scenarios, such as highly irregular masks, extreme semantic gaps, or out-of-distribution content. A more detailed discussion of limitations would strengthen the paper’s practical relevance.
3. Although the paper compares against several strong baselines, it omits recent zero-shot inpainting or image-editing methods such as “Inpaint Anything,” “DiffEdit,” or “pix2pix-zero.” Including these would provide a more comprehensive evaluation of DING’s relative performance and efficiency.

**Questions:**

1. Could the authors provide a more systematic analysis of where DING tends to fail? For instance, how does it perform on very large or very small masked regions, or in cases with significant semantic discontinuities? Does it ever overfit to the observed pixels at the expense of realism or coherence?
2. Are the authors able to include comparisons with recent methods such as “Inpaint Anything,” “DiffEdit,” or other zero-shot image-editing approaches? If not, could they at least provide a qualitative or quantitative discussion of how DING might compare in terms of performance and efficiency?
3. The method is currently tailored for inpainting. Could the authors comment on the feasibility of extending the decoupled guidance framework to other inverse problems (e.g., super-resolution, deblurring) while maintaining similar efficiency gains?

---

> ### Author Response · Authors · 2025-11-20
>
> Dear Reviewer,
>
> We would like to thank you for your detailed feedback on our paper. Below we address your remaining concerns.
> > The manuscript lacks a systematic analysis of failure cases or challenging scenarios, such as highly irregular masks, extreme semantic gaps, or out-of-distribution content. A more detailed discussion of limitations would strengthen the paper’s practical relevance.
>
> We thank you for this suggestion. We have updated the appendix of the paper by including explicit examples of failure cases when the mask is too large for example and the prompt is not precise enough. In such cases the method still outputs coherent images but the background may lose consistency. See Appendix A.4 in the updated manuscript.
>
> > Although the paper compares against several strong baselines, it omits recent zero-shot inpainting or image-editing methods such as “Inpaint Anything,” “DiffEdit,” or “pix2pix-zero.” Including these would provide a more comprehensive evaluation of DING’s relative performance and efficiency.
>
> Thank you for the suggestions. Regarding Inpaint Anything [1], we note that it primarily proposes a segmentation–inpainting pipeline built on top of a Stable Diffusion model fine-tuned for inpainting. Since our evaluation already includes comparisons with the fine-tuned SD3 inpainting model, we believe this effectively covers the same setting. Regarding pix2pix-zero, we note that although it performs zero-shot editing using a pre-trained diffusion model, it is not designed for mask-based inpainting or localized editing. The method operates at the global image level, applying semantic translations (e.g., “cat → dog”) without any spatial mask input to restrict edits. As shown in the paper’s visual examples, this often leads to changes spreading beyond the intended regions, altering background or contextual elements outside the area of interest. In contrast, our inpainting setting explicitly constrains edits to masked regions, preserving the rest of the image exactly. Therefore, pix2pix-zero is not directly comparable to our mask-based framework. Finally, for DiffEdit we note that it provides a pipeline where the mask is deduced from the reference image and the target prompt, and then in the reverse steps it performs reverse diffusion steps akin to Blended Diffusion [3]. In our setting we assume access to the mask and do not need to deduce it, so we proceeded to simply compare against latent Blended Diffusion.
> We have included two more baselines requested by reviewer S6KU, DiffPIR and DDNM. The results can be found in the updated version of the paper, Tables 2 and 3 in the experiments section of the main paper. Overall, the conclusion from our paper doesn’t change and our method still provides the best trade-off. We would like to insist that our paper now includes 11 baselines in total.
>
> > While the decoupled likelihood surrogate is a clear improvement in efficiency and simplicity, it can be viewed as an incremental extension of existing VJP-free or mixture-based guidance methods (e.g., Wu et al., 2023; Janati et al., 2024/2025a). The paper could more explicitly articulate its conceptual departure from these prior works to better highlight its foundational contribution.
>
> We respectfully disagree that our method is an incremental extension of existing VJP-free or mixture-based guidance approaches. The works cited by the reviewer remain VJP-based, whereas our method is fundamentally VJP-free. While our method and these works also consider the surrogate to the posterior transition in equation 3.1, here we define a different type of transition involving a likelihood surrogate not considered yet in the literature so far. Thus, our method shares the same objective as the methods cited by the reviewer, i.e. approximate the posterior transition, but do so in a very different way. Concerning the VJP-free methods considered in the paper, and against which we compare, we explicitly highlight their differences from our approach in the appendix through a detailed comparison of the update equations. To further clarify the distinction of our method from existing works, we have added a new paragraph in Appendix A.2 highlighting the differences with other existing methods. These additions are included in the updated version of the paper for the reviewer’s reference.

---

> > ### Author Response · Authors · 2025-11-20
> >
> > > The method is currently tailored for inpainting. Could the authors comment on the feasibility of extending the decoupled guidance framework to other inverse problems (e.g., super-resolution, deblurring) while maintaining similar efficiency gains?
> >
> > The main challenge in extending our method to other inverse problems lies in handling the forward operator. In the case of inpainting, the mask can be naturally transferred into the latent space, resulting in a simple linear operator. This property is key to DING’s efficiency and accuracy. For other inverse problems, such as super-resolution or deblurring, this transfer is not straightforward; there is no direct way to express the operator in latent space without training a dedicated latent operator (as explored, for example, in [2]). Extending our framework in a training-free manner would therefore require operating in pixel space and differentiating through the decoder. In this setting, our likelihood surrogate can still be applied by approximating the conditional transition on $z_t$ (as described before Eq. 3.3) via Gaussian variational inference. However, we found that this approach is sensitive to the parameterization of the Gaussian approximation. If not carefully designed, it tends to produce overly smooth reconstructions. Addressing this limitation and stabilizing the extension to general inverse problems is part of our ongoing work.
> >
> > **Further paper updates**: Beyond the added baselines, we have also extended the comparison with the SD3 inpainting baseline under higher NFE budgets, where our approach still outperforms it. We have also introduced a new section discussing the limitations of our method (Section A.4). All of the modifications are highlighted in green in the updated PDF on openreview.
> > We again thank the reviewer for their constructive feedback and valuable suggestions. We remain open to any further questions or suggestions the reviewer may have.
> >
> > [1] Yu, T., Feng, R., Feng, R., Liu, J., Jin, X., Zeng, W. and Chen, Z., 2023. Inpaint anything: Segment anything meets image inpainting.
> > [2] Raphaeli, R., Man, S. and Elad, M., 2025. Silo: Solving inverse problems with latent operators.
> > [3] Avrahami, O., Fried, O. and Lischinski, D., 2023. Blended latent diffusion.

---

### Meta-Review · Area_Chair_t8ch · 2026-01-07

**Summary:**

This paper proposes a new heuristic to improve the efficiency of zero-shot inpainting methods based on pretrained diffusion models. The approach is simple yet effective, delivering meaningful gains across numerous experiments compared to existing methods. Reviewers appreciated the paper’s contribution, clarity, and experimental validation. During the rebuttal phase, the authors responded to  reviewers' concerns, including requests for deeper theoretical justification, broader comparisons with other methods, and potential generalization to inverse problems beyond inpainting. While some concerns, such as providing additional theoretical insights, may not have been fully resolved, I believe the current version makes a valuable contribution to the field. Therefore, I recommend acceptance of the paper.

**Reviewer Concerns:**

In their rebuttal, the authors strengthened the paper’s experimental validation by addressing concerns raised by reviewers SWvq, PHGS, and S6KU. Specifically, they added additional experiments, clarified experimental settings, and expanded comparisons with existing approaches by including more baseline algorithms. The authors also attempted to provide deeper theoretical insights and justification for their methods as requested, however, this aspect was not fully resolved.

**Reviewer Scores:**

All reviewers initially gave a score of 6 and maintained it. Reviewers SWvq and MmE3 would likely have increased their scores after the rebuttal. Given that most concerns from the remaining reviewers were addressed, it is reasonable to assume they would either keep their scores or raise them following the rebuttal.

---

### Decision · Program_Chairs · 2026-01-26

Accept (Poster)